# N2M: Bridging Navigation and Manipulation by Learning Pose Preference from Rollout

## Abstract

In mobile manipulation, the manipulation policy has strong preferences for initial poses where it is executed. However, the navigation module focuses solely on reaching the task area, without considering which initial pose is preferable for downstream manipulation. We identify this critical, yet highly overlooked problem and introduce N2M, a strongly practical solution that guides the robot to a preferable initial pose after reaching the task area, thereby substantially improving task success rates. N2M features five key advantages: (1) reliance solely on ego-centric observation without requiring global or historical information; (2) real-time adaptation to environmental changes; (3) reliable prediction with high viewpoint robustness; (4) broad applicability across diverse tasks, manipulation policies, and robot hardware; and (5) remarkable data efficiency and generalizability. N2M demonstrates state-of-the-art performance compared to prior methods showing 3% to 54% performance improvement compared to reachability-based methods and 24% to 55% performance improvement compared to the only existing policy-aware alternative in *PnPCounterToCab* and *CloseDrawer* tasks respectively. Furthermore, in the *Toybox Handover* task, N2M provides reliable predictions even in unseen environments with only 15 data samples, showing remarkable data efficiency and generalizability. Anonymized project website: https://nav2manip.github.io

## 1 Introduction

Mobile manipulators, which integrate mobility and environmental interaction capabilities, hold significant promise for a wide range of real-world applications. By leveraging scene understanding Rana et al. (2023); Hughes et al. (2022); Rosinol et al. (2020) and navigation modules Zheng et al. (2025); Chai et al. (2024); Chang et al. (2023), these robots can reach the task area based on the task descriptions, and subsequently accomplish the task by executing pre-trained manipulation policies Fu et al. (2024); Chi et al. (2024); Black et al. (2024).

However, existing works mainly focus on enhancing navigation and manipulation independently, while not giving sufficient attention to the interplay between them. In this paper, we identify an inherent misalignment between navigation and manipulation, which significantly reduces the task success rate. Specifically, due to factors such as joint limitation and training data distribution, the performance of the manipulation policy is sensitive to the initial pose from which execution begins. Meanwhile, navigation merely focuses on guiding the robot to task areas without considering which initial pose is preferable for executing the manipulation policy.

The most direct solution would be to develop an end-to-end model handling both navigation and manipulation Yang et al. (2024), thereby avoiding challenges in inter-module coordination. However, due to the inherent complexity of both navigation and manipulation, the design, training, and data collection for such end-to-end models remain an open problem. An alternative approach within modular frameworks is to enhance the robustness of the manipulation policy. However, visuomotor policies are sensitive to viewpoint changes Heo et al. (2023), necessitating data collection from various initial poses throughout the task area Gu et al. (2022), which is costly.

In this paper, we propose a simple but effective transition module, named N2M (Navigation-to-Manipulation), serving as a bridge between navigation and manipulation. As depicted in Fig. 1, after reaching the task area, the robot is transferred from the end pose of navigation to an initial pose that is preferable for executing the manipulation policy, thereby improving the task success rate.

Figure 1: System overview. The transition process from the navigation end pose to manipulation initial pose.

We identify five fundamental challenges for bridging navigation and manipulation, and propose our corresponding solutions.

**Adaptability to non-static environments**. The environments are typically non-static, requiring predictions to adapt to environmental changes. To support this, N2M predicts the preferable initial pose from the ego-centric RGB point cloud with a single forward pass. This efficient design enables N2M to generate real-time predictions in dynamic environments, as demonstrated in Section 5.2.

**Multi-modality of preferable initial poses**. Multiple preferable initial poses may exist within the task area. Consequently, predicting a single pose is insufficient, as it can cause the model to learn an interpolation between viable poses Bishop (1994), which may not be preferable to execute manipulation policies. To address this multi-modality, N2M predicts the distribution of preferable initial poses, which is represented with a Gaussian Mixture Model (GMM) Bahl et al. (2023).

**Criterion of preferable initial poses**. Manipulation performance depends on multiple factors: policy architecture, training data distribution, robot configuration, task, and environment. Rather than attempting to model these complex relationships, we directly evaluate the pose through policy rollouts. During data collection, we position the robot at various poses and execute the manipulation policy, and successful execution indicates a preferable initial pose. Learning initial pose preferences directly from policy rollouts ensures that N2M's predictions align with the policy's actual performance while simultaneously enabling broad applicability across diverse policies, tasks, and robot hardware, as shown in Sections 4.2, 5.1, and 5.2.

**Viewpoint Robustness**. Since the robot navigation end poses can be anywhere within the task area, N2M needs to provide reliable predictions at various viewpoints. To achieve this, we augment N2M's training data from multiple viewpoints. Experiments in Sections 4 and 5 demonstrate that N2M reliably predicts preferable initial poses across the whole task area. Interestingly, we note that our proposed data augmentation approach also significantly improves data efficiency and generalizability. We will further analyze the reason behind these benefits in Section 6.

**Data Efficiency**. Collecting rollouts requires substantial time and human effort, as each rollout must be monitored and manually labeled with success or failure. We incorporate two main strategies to make N2M data-efficient: First, we design the module to directly predict the initial pose distribution, rather than low-level action Lee et al. (2019); Second, we augment the dataset through viewpoint rendering to increase its coverage and diversity. In Sections 4.3, 4.4, and 5, we demonstrate that N2M has remarkable data efficiency and generalizability.

Our contributions are as follows:

First, we identify and analyze the critical misalignment between navigation and manipulation modules. With experiments, we show that this misalignment stems from the manipulation policy's extreme sensitivity to its initial pose, a factor largely overlooked by prior work.

Second, we propose N2M, a novel method that directly addresses this problem. N2M stands out as a highly practical solution, demonstrating broad applicability, real-time performance, viewpoint robustness, remarkable data efficiency, and generalizability.

Finally, we conduct extensive experiments validating the effectiveness and showing state-of-the-art performance of our proposed N2M module across various settings and will release our code to facilitate community exploration.

## 2 RELATED WORK

### 2.1 NAVIGATION

Model-based navigation has advanced significantly over the past few decades, enabling mobile manipulation robots to navigate without collisions in unstructured environments Zheng et al. (2025). The users typically need to explicitly provide the coordinates of the navigation target, which can be obtained by constructing the semantic map Rosinol et al. (2020) or scene graph Hughes et al. (2022); Bavle et al. (2023) that associates semantic information with location Rana et al. (2023). Additionally, RL-based object navigation Ye & Yang (2021), or zero-shot navigation based on Large Language Models (LLMs) Yao et al. (2024) and Vision-Language Models (VLMs) Zhang et al. (2024b), can also be integrated into the mobile manipulation system Chen et al. (2023); Kuang et al. (2024). However, these systems can only determine navigation targets through heuristic rules Chang et al. (2023); Wang et al. (2023); Liu et al. (2024), such as requiring the robot to face the target object or remain within a specified radius of it. Such heuristics lack the connection to subsequent manipulation policy, often resulting in suboptimal positioning and failures in manipulation.

### 2.2 MANIPULATION

Data-driven approaches have demonstrated their advantages in complex and dexterous manipulation tasks. Through experience Mandlekar et al. (2020); Zhang et al. (2024a) or human demonstrations Zhao et al. (2023b); Chi et al. (2023), robots can learn manipulation policies. However, due to hardware configuration Gadre et al. (2022), environmental factors Abdelrahman et al. (2024), and the distribution of training data Gao et al. (2024), executing pre-trained policies at different initial poses within the task area yields significantly different success rates. One possible solution is to enhance the robustness of policies to initial poses. $\pi_0$ Black et al. (2024) improves generalizability by training the policy on large-scale data collected throughout the task area, building upon pre-trained VLM models. Stem-OB Hu et al. (2024) utilizes pre-trained image diffusion models to suppress low-level visual differences while maintaining high-level scene structures. Alternatively, we propose N2M, which effectively predicts the distribution of initial poses preferred by the manipulation policy, without requiring extensive data collection for the robustness of manipulation policies.

### 2.3 BRIDGING NAVIGATION AND MANIPULATION

The importance of selecting appropriate initial poses for manipulation with mobile robots has long been recognized. Pioneering works addressed this problem by calculating the Inverse Reachability Map (IRM) Vahrenkamp et al. (2013); Jauhri et al. (2022), defining preferable initial poses as placements where the target object is guaranteed to be reachable. While this reachability-based criterion is sufficient for planner-based policies, it falls short for data-driven policies.

More recent line of works MoManipVLA Wu et al. (2025b) and MoTo Wu et al. (2025a) aim to extend off-the-shelf fixed-base manipulation policies with mobile capability. However, similar to the above-mentioned IRM methods, they rely solely on geometric analysis to determine the initial base pose, thereby critically omitting the manipulation policy's sensitivity and preferences. This reliance on geometric methods makes them ineffective for data-driven manipulation policies.

Mobi-$\pi$ Yang et al. (2025), a concurrent and closest existing work to ours, is currently the only policy-aware method for determining the initial base poses. However, Mobi-$\pi$ suffers from significant practical issues. First, Mobi-$\pi$'s reliance on the policy's original training data makes the method incompatible with state-of-the-art methods that are often pre-trained on vast datasets Intelligence et al. (2025); Team et al. (2025) or refined through continuous RL feedback Amin et al. (2025); Lei et al. (2025). Second, Mobi-$\pi$ incurs a significantly large inference time because it calculates scores for a given base pose rather than directly predicting the desired pose, necessitating extensive search, iterative sampling, and 3DGS scene reconstruction. This low efficiency also inherently restricts its application to static environments. In contrast, N2M is highly practical since it treats the manipulation policy as a black box, learning directly from rollouts without making any assumptions. Moreover, its fast inference enables immediate adaptation to dynamic environmental changes, providing a critical advantage.

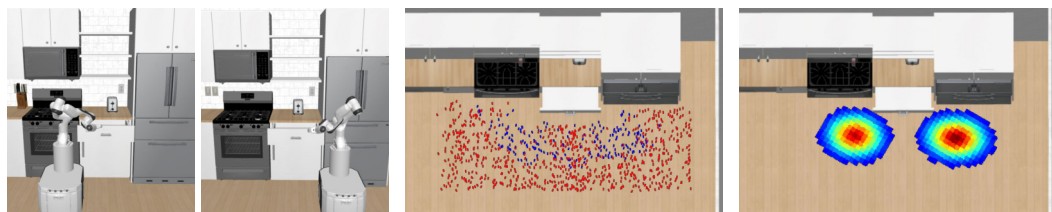

(a) Robocasa Close Drawer Task     (b) Rollout Results     (c) Predicted Preferable Initial Pose Distribution

Figure 2: (a) Preferable initial poses of the *Close Drawer* task are inherently multi-modal, as the manipulation policy is learned to close drawers from both sides. (b) visualizes preferable initial poses from successful rollouts (blue), which shows multi-modality. They are distributed on both sides, with multiple valid poses per side. (c) With GMM, we effectively model this multi-modal nature of preferable initial poses.

## 3 METHODOLOGY

### 3.1 N2M MODULE OVERVIEW

As illustrated in Fig. 1, our proposed N2M module consists of four steps. First, at the navigation end pose, we capture an RGB point cloud with the RGB-D camera mounted on the robot. Second, our N2M network predicts the distribution of the preferable initial poses from the captured point cloud. Third, a collision-free pose is sampled from the predicted distribution. Finally, the robot navigates to the selected initial pose to execute the pre-trained manipulation policy.

### 3.2 NETWORK ARCHITECTURE

As the core component of N2M, our network outputs the distribution of initial poses that is preferable for executing manipulation policies. We utilize an RGB point cloud captured from the RGB-D camera mounted on the robot as the input of the network. This design enhances practicality by relying solely on onboard sensors, without any global or historical information during inference.

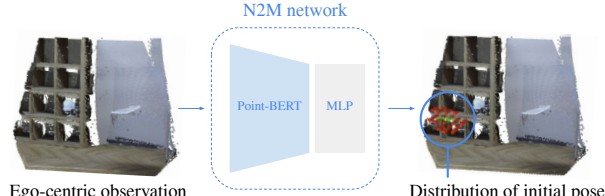

Figure 3: N2M network takes ego-centric RGB point clouds to predict the distribution of preferable initial poses.

To effectively capture the multi-modal nature of preferable initial poses within a task area, as shown in Fig. 2, we model the distribution of preferable initial pose $p_\pi$ with GMM:

$$P(p) = \sum_{k=1}^{K} \alpha_k \mathcal{N}(p|\mu_k, \Sigma_k), \tag{1}$$

where $K$ denotes the number of Gaussian kernels, $\alpha_k$ represents the weight of the $k$-th kernel, and $\mathcal{N}(p|\mu_k, \Sigma_k)$ signifies the Gaussian distribution with mean $\mu_k$ and covariance matrix $\Sigma_k$.

Our N2M network, $f_\theta$, predicts the parameters $\{\alpha_k, \mu_k, \Sigma_k\}_{k=1}^{K}$ using RGB point cloud observation $o$, captured by an onboard RGB-D camera. As illustrated in Fig. 3, the point cloud is encoded by Point-BERT Yu et al. (2022) into a fixed-length latent vector, which then passes through a multi-layer perceptron (MLP) to generate the parameters for each Gaussian kernel of the GMM.

The network is trained by optimizing a negative log-likelihood loss function that maximizes the probability of preferable initial poses,

$$L(\theta) = \sum_{(o_i, p_i) \in D} -\log P_{f_\theta(o_i)}(p_i), \tag{2}$$

where $D$ denotes the dataset consisting of observation–pose pairs, and $(o_i, p_i)$ represents the $i^{\text{th}}$ element in the dataset. We fine-tune the pre-trained Point-BERT along with MLP layers as we empirically find that this leads to better performance. Training details can be found in Appendix A.

### 3.3 DATA PREPARATION

Preparing training data for the N2M network involves two steps: collecting the raw dataset $R$ and augmenting it to create the training dataset $D$.

#### 3.3.1 RAW DATA COLLECTION

The raw dataset $R$ consists of entries $(S_i, p_{\pi,i})$, where $S_i$ represents a local scene reconstruction and $p_{\pi,i}$ denotes a preferable initial pose for manipulation policy execution, as illustrated in Fig. 4(a).

For each entry, the collection process proceeds as follows:

1. Multiple RGB point cloud frames are captured and stitched together to reconstruct the local task area $S_i$, with their relative poses determined through odometry Mohamed et al. (2019) or point cloud registration Huang et al. (2021).

2. A pose within the task area is selected for policy rollout. If the manipulation policy $\pi$ successfully completes the task, this pose is recorded as $p_{\pi,i}$ for the current scene $S_i$. The scene is then randomly reset for the next rollout.

Note that $p_{\pi,i}$ and $S_i$ must share the same coordinate frame. However, the specific choice of reference frame does not matter, as we will transform the coordinate during both training and inference to the body coordinate of the robot.

#### 3.3.2 DATA AUGMENTATION

N2M is designed to predict the distribution of $p_\pi$ based on ego-centric observations. Since the navigation end pose $p_{\mathrm{nav}}$ can be anywhere within the task area, we apply data augmentation to enhance N2M's robustness to viewpoint variations.

As shown in Fig 4(b), for each collected scene-pose pair $(S_i, p_{\pi,i})$, we first uniformly sample $M$ different viewpoints within the task area. We then filter out viewpoints that either collide with the scene or from which the object is not visible. For each verified viewpoint, $v$, we render point cloud $o_i^v$ by projecting points from $S_i$ to the viewpoint using the intrinsics

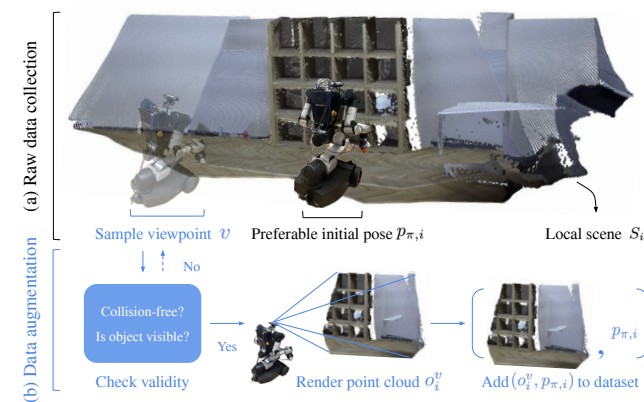

Figure 4: Data preparation process: (a) Raw data collection, showing scene $S_i$ and preferable pose $p_{\pi,i}$; (b) Data augmentation, rendering scene from diverse viewpoints.

of the RGB-D camera mounted on the robot. Note that the preferable initial pose $p_{\pi,i}$ for a given scene $S_i$ remains invariant across viewpoints. Therefore, all rendered observations $o_i^v$ from the same scene share the same label $p_{\pi,i}$, and each pair $(o_i^v, p_{\pi,i})$ is added to the training dataset $D$.

Ablation studies can be found in Appendix F, which will justify architecture and parameter choices.

## 4 SIMULATION EXPERIMENT

### 4.1 EXPERIMENT SETTING

We conduct our experiments in RoboCasa Nasiriany et al. (2024), a simulation platform that offers diverse manipulation tasks with pre-collected demonstrations for training manipulation policies. We introduce three baselines to demonstrate N2M's effectiveness:

(1) Reachability: Most existing work determine the robot pose by computing reachability, which ensures that the end-effector of the mobile manipulator can reach the target object while maintaining collision-free configurations. This baseline represents the work of MoTo, MoManipVLA, IRM.

(2) Mobi-$\pi$: This concurrent work computes similarity scores between observations from different viewpoints in the task scene and the training data. It employs Bayesian optimization to sample online from a pre-established 3D Gaussian Splatting scene representation to determine the base position.

(3) Oracle, evaluating manipulation policy at fixed pose, the same pre-defined pose used during demonstration collection in RoboCasa. As it reflects in-distribution performance, the oracle is expected to perform well.

Detailed experiment settings are provided in Appendix B.

## 4.2 Broad Applicability of N2M

Since N2M doesn't rely on any assumptions about tasks, policies, and robot hardware, it has a broad applicability. To validate this, we select four predefined tasks and three policy designs in RoboCasa. We train multiple N2M modules using 5 to 70 successful rollouts and select the best-performing model for each setting. We report the averaged success rate from 300 trials in Fig. 5.

Across all settings, N2M consistently outperforms the reachability baseline, demonstrating that naive integration of navigation and manipulation, without accounting for the policy's preference, leads to poor performance. This emphasizes the necessity of N2M that can effectively bridge navigation and manipulation. Second, the success rate with our N2M module is comparable to the oracle baseline, indicating that N2M can reliably estimate preferable initial poses across various settings.

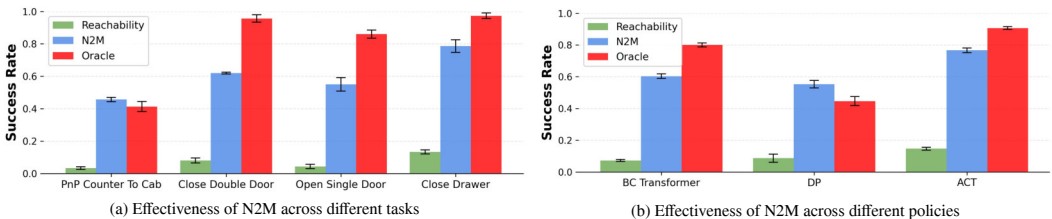

(a) Effectiveness of N2M across different tasks    (b) Effectiveness of N2M across different policies

Figure 5: Performance across different (a) tasks and (b) policies

Notably, for the *PnPCounterToCab* task in Fig 5(a) and DP in Fig 5(b), N2M outperforms the oracle baseline. This is especially remarkable, as it demonstrates that the policy's preference does not necessarily align with the distribution from training data. N2M module, trained directly from policy rollouts, effectively captures these preferences, achieving superior performance compared to the oracle baseline. This finding wouldn't have been possible with similarity-based in-distribution estimation methods, highlighting the importance of learning from rollouts that reflect the actual behavior and preference of the manipulation policy.

## 4.3 Data Efficiency of N2M

This experiment shows the data efficiency of N2M. We choose the *PnPCounterToCab* task to demonstrate this feature.

We evaluate the averaged success rate of N2M modules trained with varying numbers of rollouts in N2M's training. In each rollout, the apple's position, color, and shape vary while the kitchen furniture remains consistent. The module is then tested in the same scene. As shown in Fig 6, the averaged success rate of the manipulation policy matches the oracle baseline with only 10 rollouts and even surpasses it with 20. Although some fluctuations indicate sensitivity to sample variations, the overall trend shows that N2M effectively captures the policy's preference with a small number of rollouts.

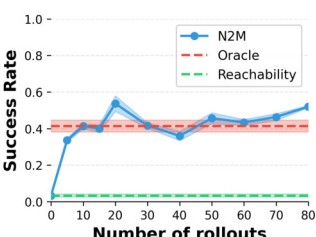

Figure 6: Data efficiency.

## 4.4 Generalizability of N2M

We evaluate the generalizability of the N2M module in the *PnPCounterToCab* task based on the number of distinct scenes used to collect successful rollouts for training. We vary the number of training scenes from 0 to 5, collecting 10 successful rollouts in each scene, resulting in a total of 0,

10, 20, 30, 40, and 50 rollouts, respectively. The trained module is then tested in an unseen scene. We design two groups of varying scenes. For the first group, as shown in Fig 7, we vary the furniture texture while keeping the kitchen layout fixed, whereas for the second group, we vary the furniture layout while keeping the furniture texture fixed.

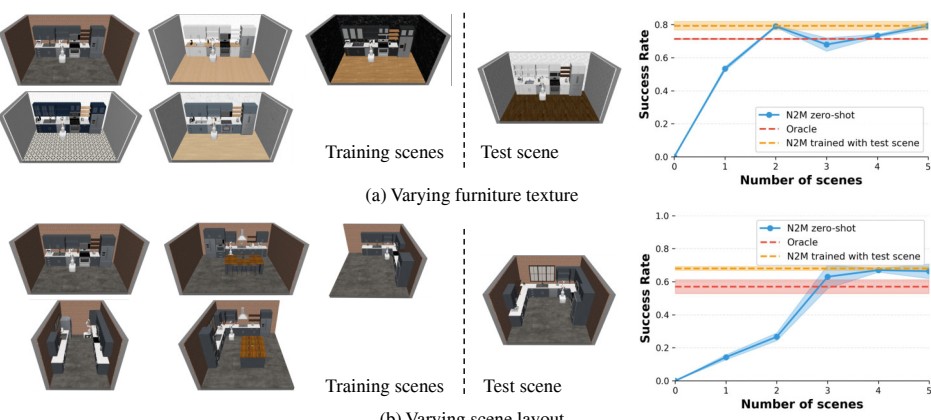

Figure 7: Experiments for testing N2M's generalizability.

The curve in Fig. 7 demonstrates that the N2M module can effectively estimate the initial pose preference of the manipulation policy even in unseen environments. As we increase the number of scenes for rollout collection, the module's performance improves accordingly, matching and even surpassing the oracle baseline. This result indicates that N2M can capture the general pattern of both the tasks and corresponding manipulation policies with a small number of scenes and apply the learned pattern in unseen scenarios.

### 4.5 BENCHMARK AND ANALYSIS

To further illustrate the distinctions between N2M and existing methods, we conducted benchmark experiments. We selected the CloseDrawer task, which is shared between Mobi-$\pi$ and our work, and tested with two different policies: BC Transformer (trained following RoboCasa's official guide and open-source data) and a diffusion policy (using Mobi-$\pi$'s open-source configuration and checkpoint). We performed 6 independent experiments, each consisting of 50 inference trails, to calculate the success rates $S_{DP}$ and $S_{BC}$ for all methods under both policies. The experiments were conducted on an NVIDIA GeForce RTX 3090 24GB GPU and an AMD EPYC 7763 64-Core Processor. Benchmark results are recorded in Table 1.

| Method | $S_{\text{DP}}$ | $S_{\text{BC}}$ | $T_{\text{human}}$ | $T_{\text{train}}$ | $T_{\text{inference}}$ |
|--------|-----------------|-----------------|--------------------|--------------------|------------------------|
| Oracle | 0.93±0.04 | 0.97±0.02 | - | - | - |
| Reachability | 0.16±0.04 | 0.15±0.03 | 0 | 0 | 0.72±0.36 s |
| Mobi-$\pi$ | 0.24±0.05 | 0.09±0.03 | $n{\times}5$ min | $n{\times}7$ min | 273.52±13.06 s |
| N2M | **0.55±0.07** | **0.56±0.04** | 12.9 min | 3.4 h | **0.07±0.03** s |

Table 1: Performance comparison with baselines. $S_{DP}$ and $S_{BC}$ denote success rates under two policies, the diffusion policy and the BC Transformer. $T_{\text{human}}$, $T_{\text{train}}$, and $T_{\text{inference}}$ represent human, training, and inference times, respectively.

**Success Rate Analysis:** Reachability-based methods show poor performance as they fail to consider the policy's pose preferences. Surprisingly, Mobi-$\pi$ doesn't perform well under either policy despite using its open-source code. We attribute this to two factors: first, observation similarity is just a heuristic of policy performance and cannot indicate the true policy performance; second, Mobi-$\pi$ uses a pretrained RGB encoder to extract RGB features, which might not be capable of distinguishing minor observation differences, as also discussed in their paper. In contrast, N2M learns pose preferences directly from actual policy rollouts, enabling more accurate prediction of preferred base positions and achieving state-of-the-art performance compared to existing methods.

**Time Cost Analysis:** Mobi-$\pi$ requires prior scene reconstruction (5min scanning + 7min 3DGS training), and Bayesian optimization during inference (273.52s each). Once the environment changes, users have to reconstruct the scene. We use $n$ to represent the number of changes. In contrast, N2M requires policy rollout collection (12.9min for 20 samples) and training (3.4 hours), but achieves rapid inference (0.07s each), enabling real-time prediction in dynamic environments.

In summary, N2M accurately predicts a high-success base pose for the manipulation task with real-time inference. In contrast, alternative approaches either suffer from low performance (reachability method) or slow inference speeds (Mobi-$\pi$). While N2M requires data collection, our proposed viewpoint augmentation technique substantially reduces the number of policy rollouts needed, making the approach more practical for real-world deployment.

## 5 REAL-WORLD EXPERIMENT

To test N2M's performance in real-world scenarios, we designed five tasks as shown in Fig 13. Detailed experiment settings can be found in Appendix C.

### 5.1 COMPREHENSIVE CASE 1

We choose the *Lamp Retrieval* task shown in Fig 13(a), and evaluate three variants of N2M module along with the reachability baseline. The difference between each variation of N2M module is how the rollouts are collected. In Fig. 8, we mark the cells where the rollouts are collected in blue, along with the success rate out of five trials. We collect one rollout for each marked cell, resulting in 3, 6, and 12 rollouts for the N2M-3 cells, N2M-half, and N2M-full variants, respectively.

As shown in Fig 8(a), the performance of the reachability baseline is notably low, indicating that naive integration between navigation and manipulation leads to poor performance. Fig. 8(b-d) shows that N2M effectively predicts preferable initial poses with only a small amount of rollouts, showcasing the data efficiency of our method. Notably, Fig. 8(b) and (c) further illustrate the generalizability of our approach: although rollouts are collected from a subset of cells, N2M can give reasonable predictions even when the lamp is placed in the cells where the rollouts are not collected.

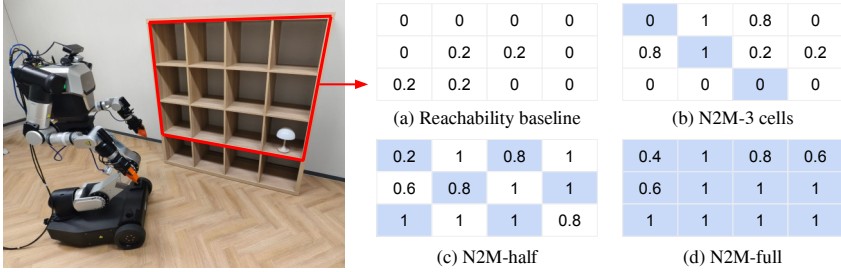

Figure 8: The *Lamp Retrieval* task with averaged success rates in each cell. The 3×4 table represents the top three rows and all four columns of the shelf. We collect one rollout per cell colored in blue to train N2M.

To test the viewpoint robustness and reliability of N2M, we demonstrate ten consecutive successful task executions, as shown in Fig. 19, with the N2M module trained using 12 rollouts. Before each execution, the lamp was randomly placed in one of the cells among the top three rows of the shelf, and the robot was randomly initialized within a $2 \times 3$ m area in front of the shelf, regarded as the navigation end pose in the task area. The robot's orientation is also randomized, but we ensure that the lamp remains visible to the RGB-D camera.

### 5.2 COMPREHENSIVE CASE 2

For the remaining four tasks shown in Fig 13(b-e), we qualitatively demonstrate N2M's remarkable data efficiency and generalizability along with its real-time performance. Note that we do not train manipulation policies for these tasks. When collecting rollouts for N2M training, we determine the preferable initial pose following our manual rule: the base is positioned approximately 0.5m away from the target object and oriented to face it.

We collect 6, 12, 6, and 15 rollouts for the tasks of *Open Microwave*, *Use Laptop*, *Push Chair*, and *Toybox Handover*, respectively, with object pose and orientation randomized within a $3 \times 6$ m room. The N2M module is then trained with these rollouts and evaluated qualitatively across various environments, including ones unseen during training. We visualize the preferable initial poses predicted by our N2M module in Fig. 9. All the images were captured based on the predictions of the N2M module. To demonstrate the adaptiveness of our method, we overlaid multiple predictions into a single image for direct comparison.

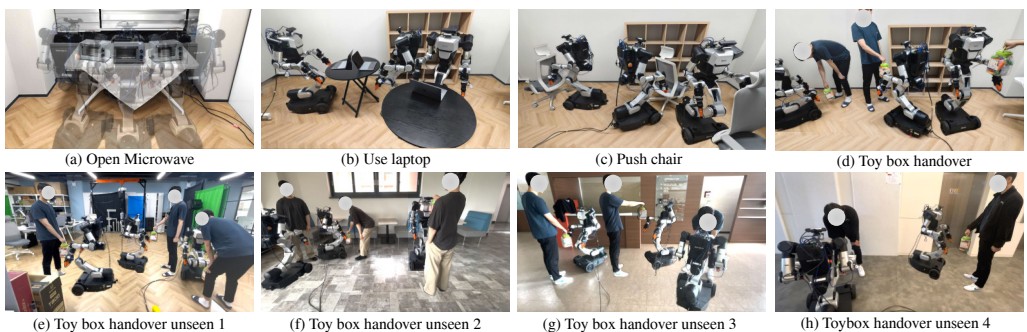

| (a) Open Microwave | (b) Use laptop | (c) Push chair | (d) Toy box handover |
| (e) Toy box handover unseen 1 | (f) Toy box handover unseen 2 | (g) Toy box handover unseen 3 | (h) Toybox handover unseen 4 |

Figure 9: Initial pose predictions among different tasks and scenes. Note that (a)-(d) are tested at unseen object placements, (e)-(h) are tested in entirely new scenes on the *Toybox Handover* task.

In Fig. 9(a-d), we evaluate the N2M module in the same environment where the rollouts used for training the N2M module are collected. The N2M module successfully predicts poses that face the object from a distance of roughly 0.5m. Especially in Fig. 9(d), the N2M module's prediction adjusts the torso height according to the height of the toybox being held by the person.

We directly deploy the same N2M module trained for the *Toybox Handover* task in four entirely unseen environments, shown in Fig. 9(e-h). In particular, we qualitatively observe that the module consistently predicts appropriate adjustments in position, orientation, and torso height based on the toybox's location and orientation. Notably, this level of generalization is achieved with only 15 rollouts collected, demonstrating N2M's remarkable data efficiency and generalizability.

Finally, we demonstrate N2M's ability to adapt predictions in real-time based on environmental changes on the *Push Chair* task shown in Fig 13(d). Figure 15 shows the predicted preferable initial poses as the chair slides across the floor. Since N2M directly predicts the preferable initial pose distribution from an ego-centric RGB point cloud with a single forward pass and without needing any historical or global information, it can generate real-time predictions in dynamic environments. In contrast, methods like Mobi-$\pi$ Yang et al. (2025) and MoTo Wu et al. (2025a), which require global scene reconstruction during inference, are less suitable for non-static environments.

## 5.3 COMPREHENSIVE CASE 3

Actually, optimizing the transition between navigation and manipulation becomes particularly crucial in sequential task scenarios. We design a multi-stage task shown in Fig. 14, which requires the robot to: (1) navigate to a table, (2) execute a pre-trained grasping policy to pick up an empty chip box, (3) navigate to a trash bin, and (4) execute a disposal policy to drop the chip box.

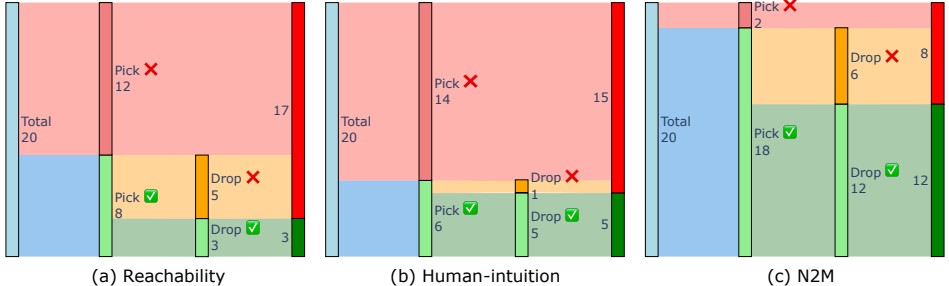

Figure 10: All the success and failure cases of the multi-stage task with different base positioning methods.

We evaluate this multi-stage task with 20 trials and under three methods: Reachability, Human-intuition, and N2M. The pose of the table, trash bin, and chip box are randomized each trial. Detailed settings are provided in Appendix C.5. The result in Fig. 10 shows our two critical insights:

First, sole reachability proves insufficient for effective manipulation. Even for humans without prior knowledge of the policy, determining the optimal base pose is challenging. Notably, we observed that after several trials, humans begin to discern the policy's positional preferences, gradually achieving higher success rates. This observation aligns with our design intuition for N2M, which similarly learns these preferences through trials, or more formally, policy rollouts.

Second, the sequential nature of this multi-stage task demonstrates how performance compounds across subtasks, highlighting the importance of optimizing base pose in each subtask. Our proposed N2M module, learning from policy rollouts, effectively captures the base preferences, substantially improving both subtask success rates and overall sequence completion. This underscores N2M's significance in multi-task mobile manipulation scenarios where sequential success is paramount.

## 6 FURTHER ANALYSIS

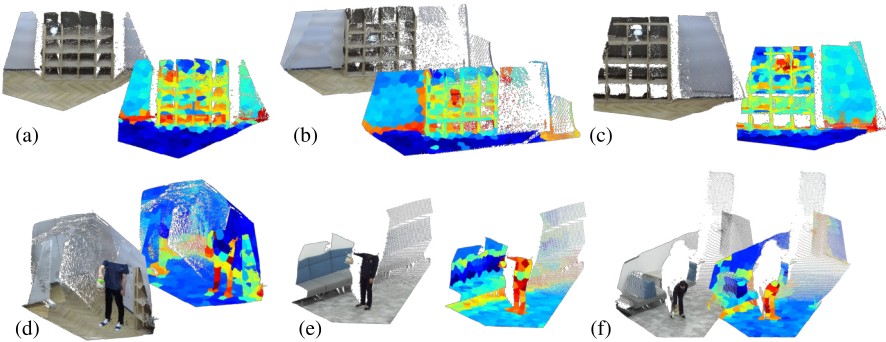

Figure 11: Learned representations from the N2M encoder. (a–c) For the *Lamp Retrieval* task, the encoder focuses on the lamp, while (d-f) for the *Toybox Handover* task, the encoder focuses on the person and the toybox. Notably, (e-f) highlights the encoder's ability to identify salient regions in unseen environments.

We further visualize the learned representation of the encoder to analyze the success of N2M. With the encoder's output features, we highlighted the region that the model focuses on. Details about the visualization can be found in Appendix G. As shown in Fig 11(a-c), in the *Lamp Retrieval* task, the model consistently focuses on the lamp, with the highlighted region shifting along with the lamp's position. In Fig 11(d-f), for the *Toybox Handover* task, the model successfully identifies the toybox and the person who is holding it. Remarkably, Fig 11(e-f) demonstrates robust generalization to unseen environments. Even though the background and the person differ from the training data, the model still identifies the toybox and the person holding it. Note again that the model is trained with 12 and 15 rollouts for *Lamp Retrieval* and *Toybox Handover* tasks, respectively, indicating that N2M learns to reliably capture salient regions in a highly data-efficient manner.

## 7 CONCLUSION AND FUTURE WORK

In this paper, we propose a simple yet effective module, N2M, that bridges the gap between navigation and manipulation by predicting preferable initial poses. We conduct extensive experiments in simulation and real world across various tasks and policies, which highlight N2M's broad applicability, remarkable data efficiency, generalizability, viewpoint robustness, and real-time performance.

To provide a comprehensive analysis, we document representative failure cases and conduct a thorough ablation study with detailed explanations in Appendix H and F.

In the future, we will focus on: (1) enabling N2M to run with only an RGB camera through monocular depth estimation and scene reconstruction to reduce hardware dependencies, and (2) incorporating failure rollouts into the learning process to prevent overestimation of initial pose preference and help the module find initial poses where policies can achieve a higher success rate.

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

APPENDIX

# A  TRAINING DETAILS

## A.1  DATA AUGMENTATION

In addition to the viewpoint augmentation described in Sec 3, we apply two further augmentations during training to improve the robustness of our module. First, we perform random rotations around the Z-axis and translations within a $1m$ radius circle on the XY-plane. Second, we uniformly down-sample the point cloud to 8,192 points, following the original Point-BERT setting.

## A.2  REGULARIZATION TERM OF LOSS FUNCTION

To better fit the distribution of preferable initial poses with GMM, we introduce three additional regularization terms. First, we maximize the entropy ($\mathcal{H}_w = -\sum_i w_i \log w_i$) of kernel weights to discourage the model from collapsing into a single mode. Second, we enforce inter-mode distance ($\mathcal{D} = \sum_{i<j}(\mu_i - \mu_j)^T \Sigma_{\mathrm{avg}}^{-1}(\mu_i - \mu_j)$) where $\Sigma_{\mathrm{avg}}$ is the average of covariance matrix) to prevent different components from converging to the same distribution. Finally, we regularize the weighted sum of entropy of each modes ($\mathcal{H}_{\mathrm{mode}} = \sum_i w_i \mathcal{H}_i$ where $\mathcal{H}_i$ is the entropy of $i^{\mathrm{th}}$ mode) to avoid overfitting by ensuring each mode does not become overly narrow. In summary, the loss function is as follows:

$$L(\theta) = \sum_{(o_i, p_i) \in D} -\log P_{f_\theta(o_i)}(p_i) - \alpha_w \mathcal{H}_w - \alpha_{\mathrm{dist}} \mathcal{D} - \alpha_{\mathrm{mode}} \mathcal{H}_{\mathrm{mode}}. \tag{3}$$

# B  DETAILED SETTINGS FOR SIMULATION EXPERIMENT

## B.1  TASK AND POLICY

We choose four tasks, as shown in Fig. 12(a-d): (a) **PnPCounterToCab**. Pick an apple from the counter and place it in the cabinet (b) **Close Double Door**. Close the cabinet doors on both the left and right sides. (c) **Open Single Door**. Open a microwave oven. (d) **Close Drawer**. Close a drawer. We got rid of the distracters during the environments. For the *PnPCounteToCab* task, we randomized the shape, color, and position of the apple. Except for the generalizability experiment in Section 4.4, all experiments were conducted in a single environment without changing the furniture texture and layout both during rollout collection and N2M inference.

In Section 4.2, we train BC Transformer Mandlekar et al. (2021) across all tasks to compare the performance across tasks. For comparison between policies, we train three different policies: BC Transformer, Diffusion Policy (DP) Chi et al. (2023), and Action Chunking with Transformers (ACT) Zhao et al. (2023a) in the *Open Single Door* task. We train each manipulation policy with 3000 demonstrations provided in RoboCasa.

To predict the distribution of the preferable initial pose of the policy, we use two kernels ($K = 2$) for the Close Drawer task as the distribution is expected to have two modes, one on each side of the drawer, and a single kernel ($K = 1$) for all other tasks.

## B.2  RANDOMIZATION CRITERION

We introduce three randomization criteria for initializing the robot pose. Note that the demonstrations provided by RoboCasa are collected from a fixed pose, and we define the randomization region based on a square centered at this reference pose.

**N2M Data collection randomization** $0.4 \times 0.4$ m square centered at the reference pose with $15°$ angular variance. Used for collecting successful rollouts to train N2M network.

**Reachability randomization** Intersection of $1 \times 1$ m square centered at the reference pose and a circle with a 1m radius centered at the target object with $30°$ angular variance. This setup captures feasible base poses for naive navigation-to-manipulation transitions based on the robot arm length.

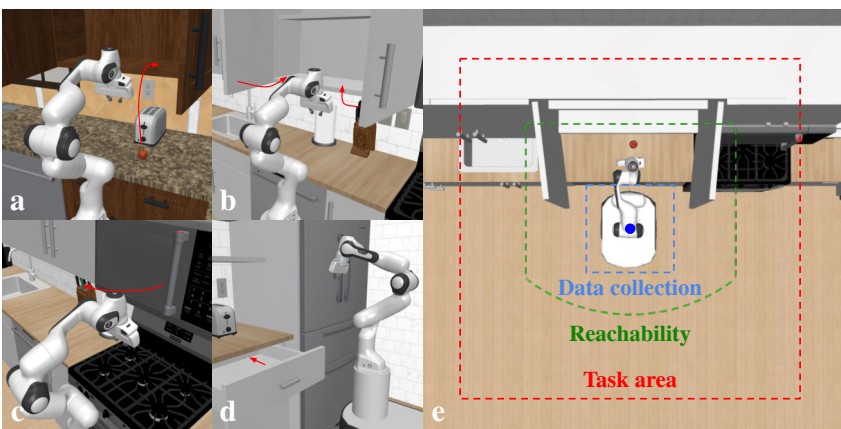

Figure 12: Task and randomization criterion in Simulation Experiment.

**Task area randomization** $2 \times 2$ m square centered at the reference pose with $30°$ angular variance. An additional constraint is imposed, requiring the target object to be visible from the given pose. The region indicates navigation end poses where we capture the RGB point cloud for N2M inference.

### B.3 ROBOT SETUP

We use a Franka Panda arm mounted on an Omron mobile base, with an additional RGB-D camera attached to the robot's wrist to capture an ego-centric point cloud. We use the ground truth depth and robot location, allowing perfect reconstruction of the point cloud.

We fix the initial joint configuration across all tasks, allowing us to decouple joint positions from the robot's base pose and predict policy preference solely in SE(2) space.

### B.4 IMPLEMENTATION OF ROBOT TRANSITION

During evaluation, after N2M prediction, we utilize MuJoCo's API to place the robot at that predicted pose for efficient simulation. We also implement a simple motion-planning algorithm for the differential-drive base to facilitate natural visualization.

## C DETAILED SETTINGS FOR REAL-WORLD EXPERIMENT

### C.1 TASK AND POLICY

For real-world scenarios, we designed five tasks, as shown in Fig. 13(a-e): (a) **Lamp Retrieval**. The lamp is randomly placed in one cell among the top 3 rows of a shelf, 12 cells in total, with variations of up to 3cm within each cell. (b) **Open Microwave**. The robot should open a microwave that is randomly placed on a white table. (c) **Use Laptop**. The robot should use a laptop that is randomly placed on a black round table, and the table is randomly placed in a room. (d) **Push chair**. The robot should push a chair that is randomly placed in a room. (e) **Toybox Handover**. The robot should take a toybox from a person randomly standing in the room and holding a toybox at varying heights.

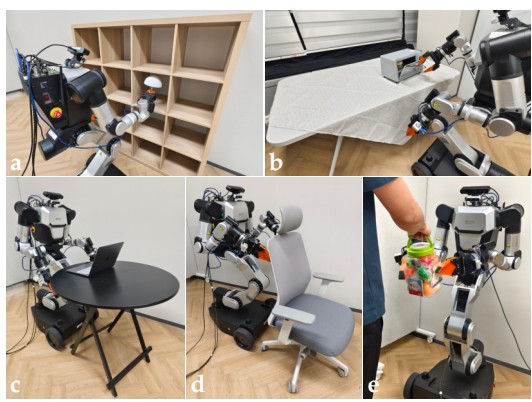

Figure 13: (a) Lamp Retrieval (b) Open Microwave (c) Use Laptop (d) Push Chair (e) Toybox Handover

For the manipulation policy in task (a), we collect 50 demonstrations from each cell with base randomized within a $0.2 \times 0.2$ m square region

with angular variance $\pm 60°$. This results in a total of 600 demonstrations, which are then used to fine-tune $\pi_0$ Black et al. (2024).

For the N2M module, we use a single kernel ($K = 1$) across all the tasks in the real world.

## C.2 RANDOMIZATION CRITERION

Specifically, for task (a), we test the N2M module with an actual policy and evaluate the success rate. For tasks (b)-(e), we test the N2M module without policy by manually labeling and gathering positive rollouts based on human-defined rules. With this setup, we demonstrate N2M's high data efficiency, generalizability, and real-time adaption to the dynamic environments.

In real-world experiments, we adopt a different randomization strategy for N2M data collection, reachability randomization, and task area randomization.

**N2M Data collection randomization** We manually pick candidates of the initial pose in the task area to collect successful rollouts, which is more efficient.

**Reachability randomization** Intersection of $0.5 \times 0.5$ m rectangular region centered 0.5m away from the object and a circle with radius 1m centered at the object with angular variance $60°$. This represents the region within the robot's reach, given the arm length is around one meter.

**Task area randomization** We utilize the whole room to randomize the base pose with additional constraint that the object should be visible. Following simulation experiment, this also indicates navigation end pose where we capture RGB point cloud for N2M inference

## C.3 ROBOT SETUP

For real real-world experiment, we employ the Rainbow Robotics RB-Y1 robot[1] platform. We use three cameras in total: a RealSense D405 camera on the right wrist of the robot, a RealSense D435, and a ZED 2i camera on the head of the robot. We use RealSense cameras for manipulation policy and the ZED 2i camera to capture the ego-centric RGB point cloud of the scene. We utilize two 2D LiDAR sensors attached to the robot base to get the odometry.

As the RB-Y1 robot offers height adjustment, we incorporate torso height into the robot's initial pose. Following the simulation setup, we fix the initial joint configuration of the robot arm, allowing us to decouple joint positions from the initial pose. As a result, the robot's initial pose is represented as a 4-dimensional vector $(x, y, \theta, h)$.

## C.4 IMPLEMENTATION OF ROBOT TRANSITION

We implement a simple motion-planning algorithm for the differential-drive base to transit the robot from the end pose of navigation to the predicted initial pose for executing the manipulation policy. Although it does not consider collisions, it is sufficient for our experiments, as motion planning is not the primary focus of our work.

## C.5 DETAILS OF COMPREHENSIVE CASE 3

**Randomization Criterion.** The trash bin and table positions are randomized within a 3m × 5m area for each trial. The chip box is randomly placed on the table surface (40cm × 80cm).

**Policy Training.** We collect 200 demonstrations for each task (picking and disposal) which are used to fine-tune the pre-trained $\pi_0$ policy.

**N2M Data Collection and Training.** For each trained manipulation policy, we conduct 20 rollouts with randomized scene configurations. We apply viewpoint augmentation with M=300 and train the N2M network for 150 epochs.

**Human-intuition Baseline.** We recruit two participants to determine robot base pose for policy execution. Participants are informed only about the task objectives but not about policy training

---

[1]https://www.rainbow-robotics.com/rby1

methodologies or data collection strategies. Robot positioning is based solely on their intuitive judgment of optimal placement.

**Reachability Baseline.** We utilize AprilTags for object pose estimation and filter out infeasible robot base pose by collision checking and IK-based reachability analysis to ensure the target object is within the manipulator's workspace.

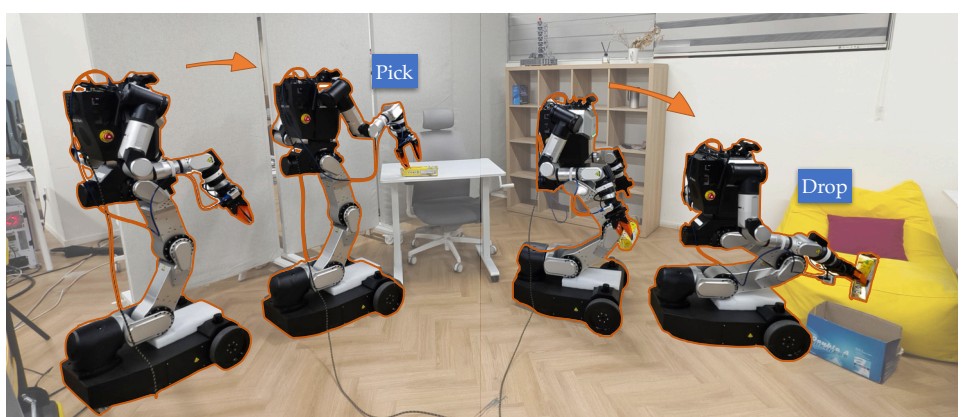

Figure 14: The illustration for the multi-stage task in comprehensive case 3.

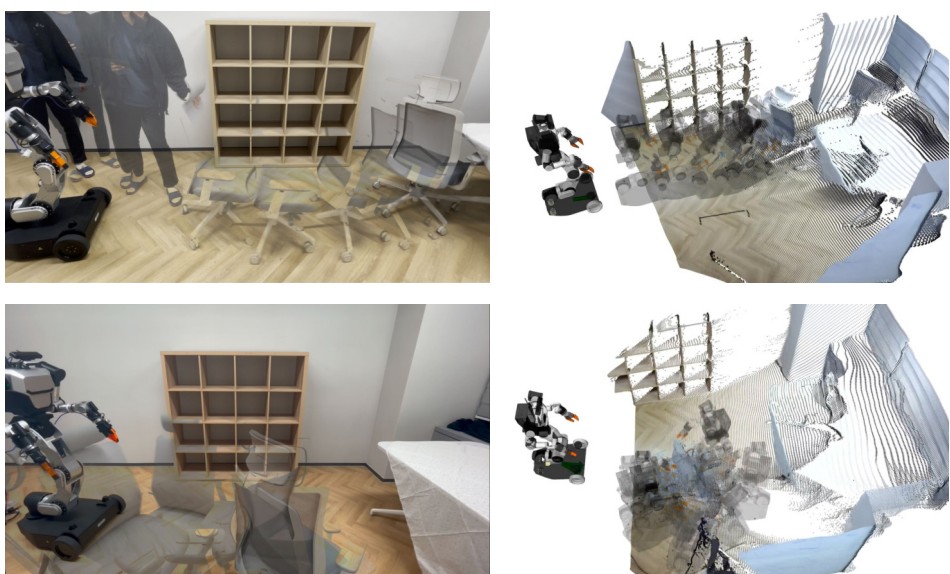

Figure 15: Real-time prediction by our proposed N2M module in the *Push Chair* task.

## D    VISUALIZATION OF VIEWPOINT ROBUSTNESS

As shown in Fig. 19, we show ten consecutive successes of the *lamp retrieval* task. Before each execution, the lamp was randomly placed in one of the cells among the top three rows of the shelf, and the robot was randomly initialized within a $2 \times 3$ m area in front of the shelf, regarded as the navigation end pose in the task area. The robot's orientation is also randomized, but we ensure that the lamp remains visible to the RGB-D camera.

## E  ADAPTABILITY TO NON-STATIC ENVIRONMENT

As shown in Fig. 15, we show two trajectories of the chair. The first row shows the result of pushing the chair in a straight line, where, as can be seen in the right image, the prediction follows the chair as it moves. The second row shows the result of spinning the chair, and we can see that the prediction rotates together with the chair. This demonstrates the adaptability to non-static scene of the N2M module that it can adapt its predictions in real-time according to changing environments.

## F  ABLATION STUDY

### F.1  EXPERIMENT SETTING

All ablation study is conducted with Pick-and-Place Counter to Cabinet task in Robocasa. We use 20 rollouts to train N2M across all settings and follow Appendix. B for other detailed inference settings.

### F.2  FURTHER ABLATION

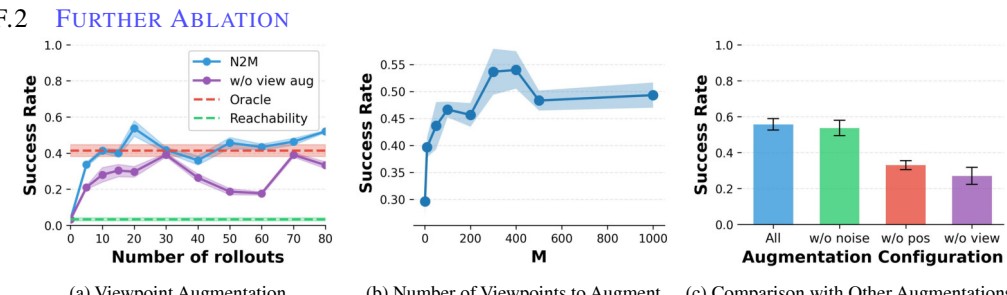

(a) Viewpoint Augmentation  (b) Number of Viewpoints to Augment  (c) Comparison with Other Augmentations

Figure 16: Ablation on (a) Viewpoint Augmentation (b) Number of views ($M$) (c) Alternative augmentations

We conducted an ablation study to analyze the impact of viewpoint augmentation on data efficiency and performance. Viewpoint augmentation is crucial for both metrics, as demonstrated in Fig. 16(a). Further analysis in Fig. 16(b) shows that performance improves with a greater number of augmented views, saturating around $M = 300$ views. Fig. 16(c) establishes viewpoint augmentation as essential for high performance. Although we compare different augmentation methods, we note that viewpoint augmentation is complementary with positional augmentation and random noise. The highest performance attained when all augmentations are combined, demonstrating the synergy of these augmentations.

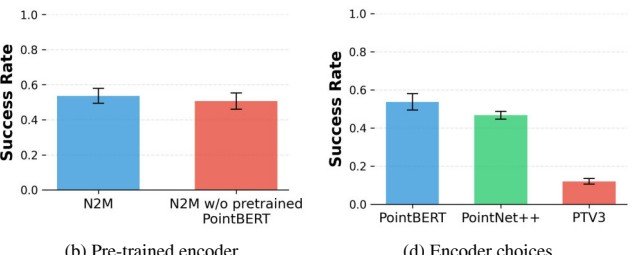

(b) Pre-trained encoder  (d) Encoder choices

Figure 17: Ablation on (a) PointBERT pretrained weight and (b) Point cloud encoder choices

We conducted a further ablation study on our point cloud encoder choice and the use of pretrained weights. Although the PointBERT Yu et al. (2022) encoder with pretrained weights achieved the highest performance (as shown in Fig. 17), the choice of encoder (PointBERT vs PointNet++ Qi et al. (2017)) and use of pretrained weights did not critically affect the overall performance of N2M.

For the low performance of PointTransformerV3, as observed in Figure 17(b), we attributed this to an inherent architectural mismatch with our specific task. First, PointTransformerV3's reliance on extracting features from relative positions ensures translation invariance. This characteristic made it inappropriate for our method, which requires predicting an absolute position. We added positional encoding based on the absolute position of the points to address this issue. An additional challenge was PointTransformerV3's output feature size, which is dependent on the voxel grid size. This led

to inconsistent output feature sizes across varying input point cloud shapes. To address this, we simply calculated the maximum value across the features to achieve size matching. Despite these algorithmic and structural compensations, we attribute the observed low performance of PointTransformerV3 to the inherent mismatch between its design purpose and the requirements of our task.

## G  LEARNED REPRESENTATIONS

To visualize where the model focuses, we calculate the similarity between the output features of each token with the feature of a learned [cls] token used in Point-BERT. As shown in Fig 11, the model learns to focus on the salient regions, which aligns with the strong performance of the N2M module. We include additional visualizations in Fig 18.

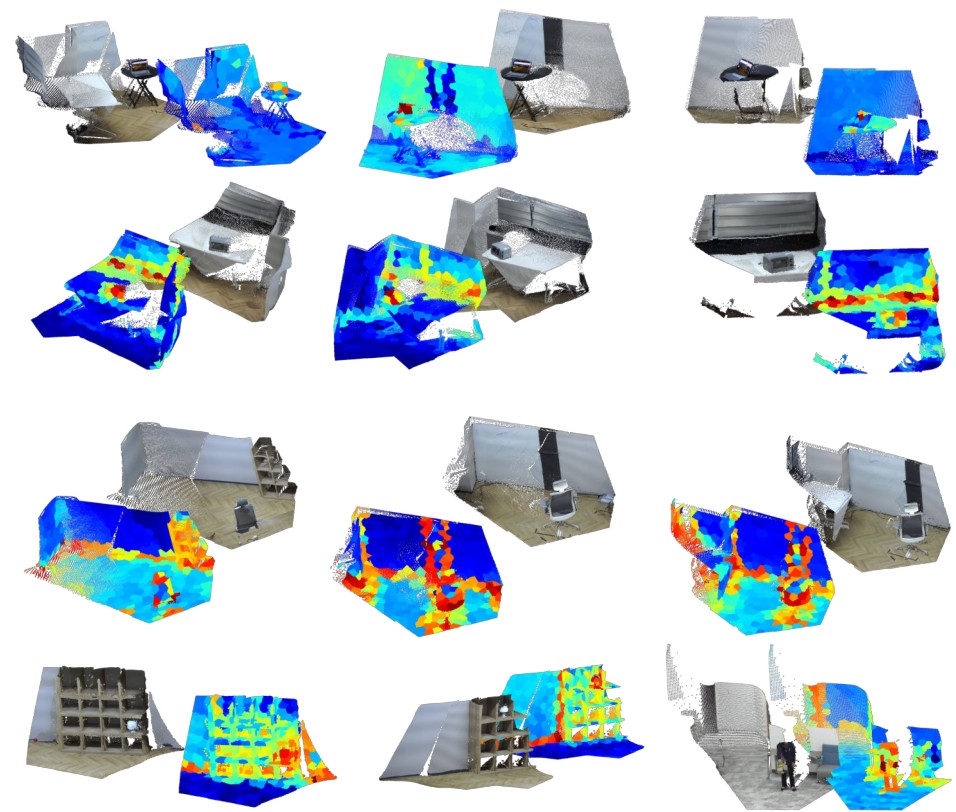

Figure 18: Visualization of learned representations of N2M.

## H  FAILURE ANALYSIS

We provide an explanation regarding factors that may affect the performance of N2M.

**Small objects:** Since we downsample the point cloud before providing it to the model, we observe that objects as small as a pen or an eraser are typically indistinguishable, leading to erroneous predictions.

**Far distance:** False predictions occur when the robot is positioned too far from the region of interest. This distance causes the region to become indistinguishable and, crucially, pushes the input outside the viewpoint sampling area used for augmentation, making it an out-of-distribution scenario.

**Noisy sensor:** Noisy estimates of the point cloud generally made the point cloud out of distribution, leading to noisy predictions. To resolve this, we used a high-performance Zed2 camera to observe high-quality point clouds.

**Manipulation policy limits:** This represents the most critical failure case observed, and it is highly relevant to the core motivation of our project. Even when our module provided a reliable prediction, we still observed manipulation failure due to the extreme sensitivity and limited capability of the underlying policy.

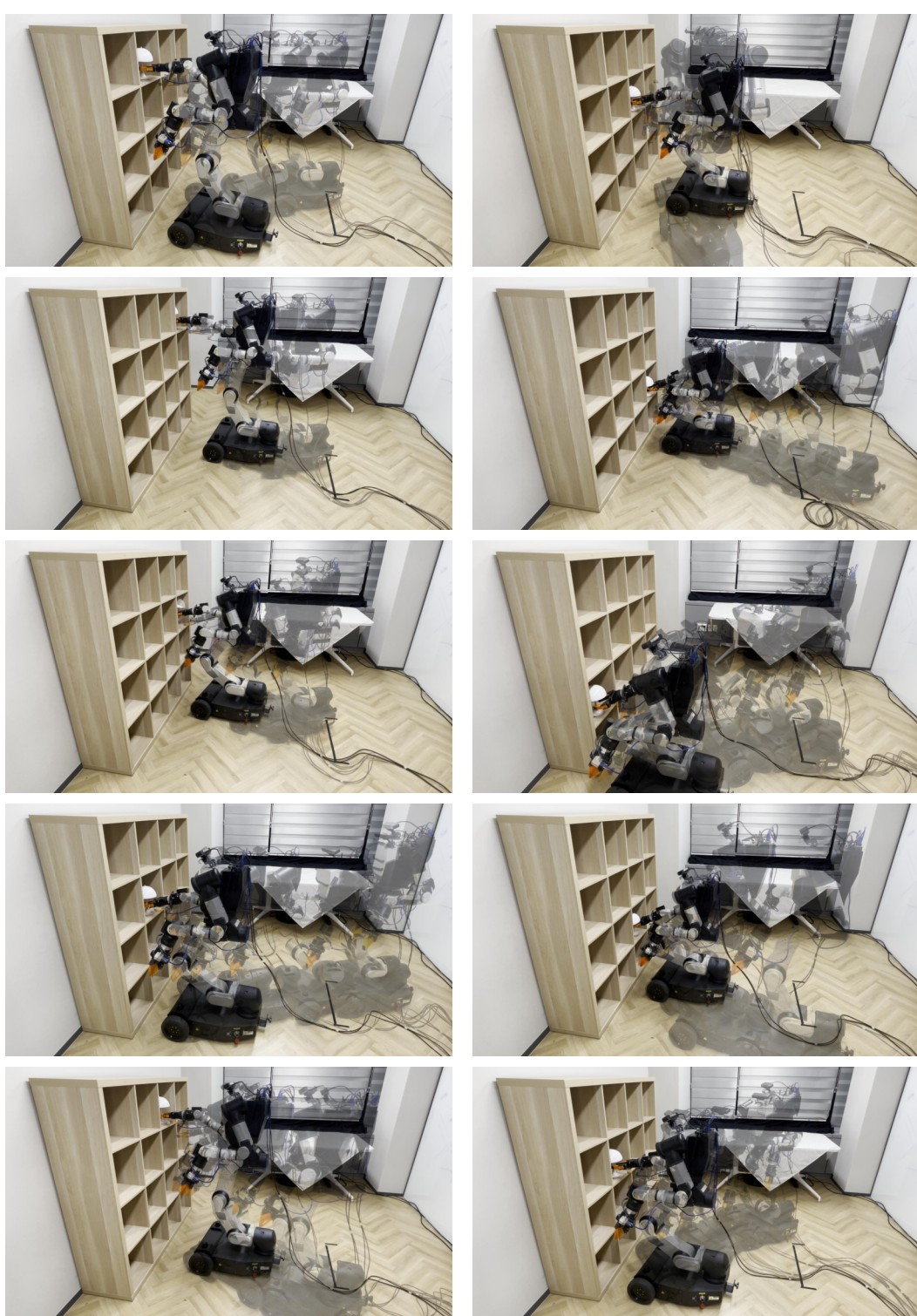

Figure 19: Ten consecutive successes of the *Lamp Retrieval* task.

