# OpenReview forum: "N2M: Bridging Navigation and Manipulation by Learning Pose Preference from Rollout"
_ICLR.cc/2026/Conference — Submitted to ICLR 2026_

### Official Review · Reviewer_ycFo · 2025-10-26

**Soundness:** 3
**Presentation:** 3
**Contribution:** 3
**Rating:** 6
**Confidence:** 5

**Summary:**

This paper identifies a critical misalignment in mobile manipulators: navigation modules deliver the robot to a task area, but not necessarily to a preferable initial pose for the manipulation policy. To bridge this gap, the authors introduce N2M, a lightweight transition module that uses only ego-centric observations to predict and guide the robot to a pose that maximizes manipulation success. Experiments in both simulation and real-world environment demonstrate the effectiveness of N2M.

**Strengths:**

1. Bridging off-the-shelf navigation and fixed-base manipulation policies with a separate fine-grained navigation model is of great application value.
2. N2M can handle dynamic scene, being viewpoint-robust and data-efficient.

**Weaknesses:**

1. More discussion is required with related works, such as MoManipVLA [1], Mobi-Pi and MoTo. All these methods empowers off-the-shelf fixed-base manipulation policy with mobile capability. A more comprehensive discussion is suggested to be placed in Introduction and Experiments.

2. The above mentioned methods are training-free, but N2M requires rollout from the manipulation policy to train. The overall data collection and training time should be reported to make a fair comparison.




[1] MoManipVLA: Transferring Vision-language-action Models for General Mobile Manipulation, CVPR 2025.

**Questions:**

After predicting the camera pose, how to efficiently optimize a path to navigation to this location?

---

> ### Author Response · Authors · 2025-11-23
> **Overview (1/n)**
>
> **Feedback from Reviewer4**: Reviewer 4 fully understands the problem addressed in this paper and believes that this work has great application value in the field of mobile manipulation. In addition, Reviewer 4 acknowledges the advantages of N2M in terms of real-time performance, viewpoint robustness, and data efficiency. The authors thank Reviewer 4 for recognizing the novelty and solid performance of our work in addressing this problem.
>
> **Concerns from Reviewer4**: Reviewer 4 astutely points out that, although the paper includes a large number of simulation and real-world experiments, these primarily demonstrate the characteristics of the proposed method itself, while lacking comparisons and further discussions with baselines other than the reachability-based approach. Furthermore, the reviewer asks how the robot navigates to the predicted base pose after the base pose has been predicted.
>
> **Reply from Authors**: The authors sincerely thank the reviewer for their professionalism and for dedicating valuable time to improving the quality of this manuscript. The authors first confirm and elaborate on the aspects that Reviewer 4 has positively acknowledged. Regarding the reviewer’s suggestion to include more discussion and comparison with related methods, we have summarized and analyzed prior work accordingly. Building on this, the authors commit to adding a detailed quantitative benchmark to significantly strengthen the paper's persuasiveness.
>
> **Next Step of Authors**:
> Furthermore, the authors will promptly provide comprehensive benchmark results, ablation studies, and supplementary real-world experiments.

---

> ### Author Response · Authors · 2025-11-23
> **Reply to Strengths (2/n)**
>
> > Bridging off-the-shelf navigation and fixed-base manipulation policies with a separate fine-grained navigation model is of great application value.
>
> Thank you for noting this. Our work is fundamentally aimed at solving the key integration problem that arises when combining existing navigation and manipulation modules. By explicitly accounting for the manipulation policy's preference, our method empowers an efficient combination that exploits the best capabilities of the manipulation policy. Our work makes three primary contributions:
>
> 1. **Problem Identification**: We identify and analyze the critical misalignment between navigation and manipulation modules. With experiments, we show that this misalignment stems from the manipulation policy's extreme sensitivity to its initial pose, a factor largely overlooked by prior work.
> 2. **A Thorough Re-paradigm**: We systematically redesign the approach to robot positioning problem, naturally overcoming limitations of previous methods:
>    - **Getting rid of assumptions**: By learning from policy rollouts rather than comparing observation similarity, N2M maintains policy awareness while eliminating dependency on policy training data.
>    - **Efficient Inference**: By employing direct distribution prediction instead of searching+iterative optimization, N2M achieves significantly faster inference and lower computational cost.
>    - **Data Efficiency Innovation**: By proposing viewpoint-based data augmentation, N2M enhances both data efficiency and viewpoint robustness, substantially reducing data collection burden for preference learning. As Reviewer 1 commented, our proposed data augmentation method has general values beyond the problem we are tackling.
> 3. **Comprehensive Validation**: We implement our method using a simple architecture (encoder+MLP+GMM) and conduct extensive experiments demonstrating N2M's new features: applicability across different robot configurations, tasks, and policy types; remarkable data efficiency and generalizability; viewpoint robustness; and real-time inference based on ego-centric observations.
>
> > N2M can handle dynamic scene, being viewpoint-robust and data-efficient.
>
> Thank you for pointing this out. These noted features are critical to making our method very practical. We summarize 5 key features, including the ones the reviewer mentioned:
>
> 1. **Broad Applicability**: Our proposed method can apply to all kinds of tasks and policy algorithms as it treats the policy as a black box when learning preferences.
> 2. **Real-time Adaptation**: Our design, utilising ego-centric observation and requiring only a single forward pass, allows our method to quickly adapt to dynamic scene changes.
> 3. **Viewpoint Robustness**: We propose viewpoint augmentation to train the N2M module, enabling it to generate reliable predictions regardless of changes in viewpoint.
> 4. **Data Efficiency**: The combination of viewpoint augmentation and simplifying the problem to direct target pose prediction (instead of low-level action sequences) enables our module to be trained in a highly data-efficient manner.
> 5. **Generalizability**: We empirically show that our module has remarkable generalizability (Sections 4.4, 5.1, and 5.2), significantly lowering adoption barriers.
>
> Evidence of this reliability is provided on our [**anonymized website (click here access)**](https://nav2manip.github.io/), where each video showcases **multiple consecutive successful predictions** in dynamic settings, a unique advantage over all existing works.

---

> ### Author Response · Authors · 2025-11-23
> **Reply to Weakness (3/n)**
>
> > More discussion is required with related works, such as MoManipVLA [1], Mobi-Pi and MoTo. All these methods empowers off-the-shelf fixed-base manipulation policy with mobile capability. A more comprehensive discussion is suggested to be placed in Introduction and Experiments.
>
> Our work can be compared with the mentioned related methods based on the following key points:
>
> 1. **Lack of Policy Awareness**: Most existing methods, including MoManipVLA and MoTo, determine the robot pose without considering the manipulation policy's preference. This often results in poor policy performance, as visuomotor manipulation policies are highly sensitive to their initial pose.
> 2. **Impracticality of Policy-Aware Alternative**: Mobi-Pi, the only concurrent policy-aware method, suffers from excessively long inference time. Its base pose prediction relies on 3D Gaussian Splatting scene reconstruction, which requires a full scan of the environment and additional online training, making it impractical for dynamic scenes and limiting its use to static environments.
>
> In contrast, our work makes two primary contributions:
>
> 1. **Problem Identification**: We identify and analyze the critical misalignment between navigation and manipulation modules. We demonstrate this misalignment stems from the manipulation policy's extreme sensitivity to its initial pose, a factor largely ignored by previous methods.
> 2. **A Practical Solution (N2M)**: We propose N2M, a novel method that directly addresses this problem. N2M stands out as a highly practical solution, demonstrating broad applicability, remarkable data efficiency, and robust generalization. It achieves real-time performance and maintains robustness across different viewpoints.
>
> As the reviewer suggested, we will provide additional information regarding the above points in the introduction and provide additional empirical comparison to justify our claim. We will be filling in the following table for comparison.
>
> | Method | N2M | Mobi-pi  | Reachability (IK-based) |
> |---|---|---|---|
> | Policy Awareness | O | O  | X |
> | Success Rate | - | -  | - |
> | Preparing Time | - | - | - |
> | Inference Time | - | - | - |
>
> > The above mentioned methods are training-free, but N2M requires rollout from the manipulation policy to train. The overall data collection and training time should be reported to make a fair comparison.
>
> We appreciate the rigor of the reviewer. We will include preparing time to ensure a fair comparison in our benchmark. Apart from the quantitative benchmark, we hope to argue that although the comparison methods are training-free, our approach demonstrates solid features in terms of both performance and practicality.
>
> 1. **MoManipVLA and MoTo**: As mentioned, these methods fall into the category of non-policy-aware approaches. While they are strong navigation papers, their focus does not account for the manipulation policy's preference, often leading to suboptimal robot base placements for the subsequent task.
> 2. **Mobi-pi**: Although this is the only existing policy-aware work, its reliance on scanning the scene, training 3D Gaussian Splats (3DGS), and optimizing camera poses results in a long, computationally expensive inference time. This significantly hinders its application, limiting its practicality to only static scenes.
> 3. **Data requirements for N2M**: While our method necessitates data collection, unlike some alternatives, our design actively focuses on reducing data requirements. This results in a very low commitment of 10–30 rollouts, requiring only approximately 20 minutes of human time, thus maintaining high practicality.

---

> ### Author Response · Authors · 2025-11-23
> **Reply to Questions (4/n)**
>
> > After predicting the camera pose, how to efficiently optimize a path to navigation to this location?
>
> Thank you for this insightful question. One straightforward way is to **reuse the existing navigation module** utilized in the preceding navigation step. Our work proposes an intermediate module specifically designed to bridge existing navigation and manipulation systems, which ensures the existence of a navigation module. Therefore, once our module predicts the refined pose ("where to go"), the problem of reaching it ("how to go") becomes easy. This intuition is a core reason why we focused on predicting the target pose, as it leverages pre-existing capabilities for efficient system integration.
>
> In our experiments, we implemented a simple SE(2) navigation algorithm to navigate the robot to the predicted robot base pose. This can also be freely replaced by any state-of-the-art model-based or data-driven goal-conditioned navigation method.

---

> ### Author Response · Authors · 2025-11-23
> **Welcoming Additional Suggestions (n/n)**
>
> We thank the reviewer for their thoughtful and positive assessment. Given the reviewer's evident deep understanding of this problem space, we would be extremely grateful for any further suggestions that could potentially enhance the clarity, rigor, or overall impact of our work. Your additional insights would be invaluable in significantly improving our work.
>
> 1. **Clarity and Interpretability**: We would greatly appreciate suggestions on any sections, including the problem definition, method description, or results, that hinder the paper's interpretability or overall flow.
> 2. **Experimental Support**: We seek the reviewer's judgment on whether all our claims are currently well-supported by the provided experimental results. If any specific claim lacks adequate empirical evidence, we are prepared to conduct additional analyses to strengthen our submission.

---

> > ### Comment · Reviewer_ycFo · 2025-11-25
> >
> > Thanks for the authors's reply. It solves most of my concerns. It is highly recommended to include detailed discussion on policy-aware and non-policy-aware baselines (MoManipVLA, MoTo, Mobi-Pi) in the revised paper. Also, the data collection cost should be reported in experiments for a fair comparison.

---

> > > ### Author Response · Authors · 2025-11-27
> > > **Result Update - Overview (1/4)**
> > >
> > > We are deeply grateful to the reviewer for their valuable time and insightful feedback that has substantially improved our paper.
> > >
> > > To address the reviewer's remaining concerns, we have conducted a comprehensive **benchmark** with relevant baselines and enhanced our manuscript with more thorough **discussions of existing methods**. Additionally, we have incorporated **multiple ablation studies** that rigorously justify our design choices, along with an analysis of **typical failure cases** and their underlying causes.
> > >
> > > We believe the newly added experiments and analysis can provide readers with a complete understanding of both the capabilities and boundaries of our approach.  Here is a summary of the new results:
> > >
> > > 1. **Demonstration of SOTA performance**: We compare our approach against the Reachability baseline (representing MoTo, MoManipVLA, and B*) and Mobi-$\pi$, the only policy-aware alternative. We evaluate both **success rates and time consumption** to provide a comprehensive and fair comparison across methods. The results show that our method significantly outperforms these baselines.
> > > 2. **Justification of design choices**: We present detailed ablation studies that validate our design choices and provide comprehensive explanations for each component of our approach.
> > > 3. **Paper Revision**: We have incorporated the newly added **benchmark and ablation studies** into the paper, including detailed analysis and discussion. Additionally, we have addressed reviewer concerns about potentially **misleading claims**. We have also rewritten the **related work** section to clarify the categorization of previous methods and their distinctions from our approach. Furthermore, we have included several typical **failure cases** to provide readers with a more complete understanding of our method's limitations.
> > >
> > > Detailed revisions can be found in our **updated paper**, marked in blue.

---

> ### Author Response · Authors · 2025-11-27
> **Result Update - Benchmark and Analysis (2/4)**
>
> To further illustrate the distinctions between N2M and existing methods, we conducted benchmark experiments. We selected the CloseDrawer task, which is the **shared task** between Mobi-$\pi$ and our work, and **tested with two different policies**: BC Transformer (trained following RoboCasa's official guide and open-source data) and a diffusion policy (using Mobi-$\pi$'s open-source configuration and checkpoint). We performed **6 independent experiments**, each consisting of 50 inference trails, to calculate the **success rates** $S_{DP}$ and $S_{BC}$ for all methods under both policies. The experiments were conducted on an NVIDIA GeForce RTX 3090 24GB GPU and an AMD EPYC 7763 64-Core Processor. Benchmark results are recorded in the following table:
>
> | **Method**      | $S_\text{DP}$    | $S_\text{BC}$    | $T_\text{human}$ | $T_\text{train}$ | $T_\text{inference}$ |
> |-----------------|------------------|------------------|------------------|------------------|----------------------|
> | **Oracle**      | 0.93 ± 0.04      | 0.97 ± 0.02      | -                | -                | -                    |
> | **Reachability**| 0.16 ± 0.04      | 0.15 ± 0.03      | 0                | 0                | 0.72 ± 0.36 s      |
> | **Mobi-$\pi$**  | 0.24 ± 0.05      | 0.09 ± 0.03      | n×5 min          | n×7 min          | 273.52 ± 13.06 s           |
> | **N2M**         | **0.55 ± 0.07**  | **0.56 ± 0.04**  | 12.9 min         | 3.4 h            | **0.07 ± 0.03 s**  |
>
> **Success Rate Analysis**:
> **Reachability**-based method shows poor performance as it fails to consider the policy's pose preferences. Surprisingly, **Mobi-$\pi$** doesn't perform well under either policy **despite using its open-source code**. We attribute this to two reasons: first, observation similarity is just a heuristic of policy performance and cannot indicate the true policy performance; second, Mobi-$\pi$ uses a pretrained RGB encoder to extract RGB features, which might not be capable of distinguishing minor observation differences, as also discussed in their paper. In contrast, **N2M** learns pose preferences directly from actual policy rollouts, enabling more accurate prediction of preferred base positions and achieving state-of-the-art performance compared to existing methods.
>
> **Time Cost Analysis**:
> **Mobi-pi** requires prior scene reconstruction (5min scanning + 7min 3DGS training), and Bayesian optimization during inference (**273.52s for each inference**). Once the environment changes, users have to reconstruct the scene. We use $n$ to represent the number of changes. In contrast, **N2M** requires policy rollout collection (12.9min for 20 samples) and training (3.4 hours), but achieves rapid inference (**0.07s for each inference**), enabling real-time prediction in dynamic environments.
>
> In summary, **N2M accurately predicts high-success base pose for manipulation task with real-time inference**. In contrast, alternative approaches either suffer from low performance (reachability method) or slow inference speeds (Mobi-$\pi$). While N2M requires data collection, our proposed viewpoint augmentation technique substantially reduces the number of policy rollouts needed, making the approach more practical for real-world deployment.

---

> ### Author Response · Authors · 2025-11-27
> **Result Update - Ablation Study (3/4)**
>
> We further provide **ablations with naive augmentations, encoder choices, the use of pretrained weights, and the number of augmented viewpoints**. Detailed explanation and graph is also provided in our revised paper Section 6 and Appendix F.
>
> **Comparison with naive augmentations**
> | **Augmentation** | **Success Rate** |
> |------------------|------------------|
> | All              | **0.557 ± 0.035**|
> | w/o noise        | 0.537 ± 0.029    |
> | w/o pos          | 0.330 ± 0.038    |
> | w/o view         | 0.270 ± 0.040    |
>
> This result establishes viewpoint augmentation as essential for high performance. Although we compare different augmentation methods, we note that viewpoint augmentation is complementary with positional augmentation and random noise. The highest performance is attained when all augmentations are combined, demonstrating the synergy of these augmentations.
>
> **Point cloud encoder choices and pretrained weights**
> | **Encoder**         | **Success Rate** |
> |---------------------|------------------|
> | PointBERT           | **0.537 ± 0.029**|
> | PointNet++          | 0.467 ± 0.022    |
> | PointTransformerV3  | 0.120 ± 0.014    |
>
> | **Model**                  | **Success Rate** |
> |----------------------------|------------------|
> | N2M                        | **0.537 ± 0.029**|
> | N2M w/o pretrained PointBERT | 0.507 ± 0.032  |
>
> Although the PointBERT encoder with pretrained weights achieved the highest performance, the choice of encoder (PointBERT vs PointNet++) and use of pretrained weights did not critically affect the overall performance of N2M.
>
> **Number of augmented viewpoints**
> | **Viewpoints** | **Success Rate** |
> |----------------|------------------|
> | M=1            | 0.297 ± 0.029    |
> | M=10           | 0.397 ± 0.019    |
> | M=50           | 0.437 ± 0.044    |
> | M=100          | 0.467 ± 0.015    |
> | M=200          | 0.457 ± 0.022    |
> | M=300          | **0.537 ± 0.029**|
> | M=400          | **0.540 ± 0.035**|
> | M=500          | 0.483 ± 0.019    |
> | M=1000         | 0.493 ± 0.023    |
>
> This result shows that performance improves with a greater number of augmented views, saturating around $M=300$ views.

---

> ### Author Response · Authors · 2025-11-27
> **Result Update - Conclusion and Request for Reconsideration (4/4)**
>
> We thank the reviewer's constructive feedback. The newly added benchmark experiment and multiple ablation studies have substantially strengthened the clarity and principles underlying this paper. **We are confident that this work addresses a novel and important problem while providing a practical, innovative solution with state-of-the-art performance**. Our key contributions include:
>
> 1. **Identifying a critical yet underestimated problem**: We demonstrate that accounting for a manipulation policy's inherent preferences and sensitivities is crucial for successful mobile manipulation. Our experiments reveal that performance decreases dramatically without this consideration (as evidenced by the Reachability baseline), underscoring the necessity of our approach.
> 2. **Proposing a highly practical method with SOTA performance**: Our N2M method not only significantly outperforms geometry-based baselines but also surpasses the only existing policy-aware alternative, Mobi-$\pi$. The extensive benchmarks validate both the effectiveness and novelty of our policy-aware approach.
> 3. **Technical contribution**: We propose a novel viewpoint augmentation method that substantially improves viewpoint robustness and data efficiency while enhancing model generalization. Moreover, it can be seen as a general 3d data processing techniques, which has significant potential applications both within and beyond the domain discussed in the paper.
>
> We are confident that our findings and solutions will advance the field of mobile manipulation. All simulation and real-world implementations (including benchmark code) will be released upon paper acceptance. In light of these comprehensive enhancements and the demonstrated significance of our work, **we respectfully invite our reviewer to reconsider the paper's contributions and impact based on the revised experimental results and manuscript**.

---

> ### Author Response · Authors · 2025-12-02
> **Result update 2 - one more real-world experiment and introduce human-intution as a new baseline**
>
> We currently have one more very interesting real-world experiment with comparison to reachability and human-intuition baselines.  We designed a multi-stage task (Section 5.3 Comprehensive Case 3) in this experiment. We believe this quantitative experiment plays a crucial role in explaining both the **SOTA performance of our method** and the **importance of the research problem** we address. The details are as follows:
>
> **Task Description:**
> We design a multi-stage task (Fig. 14), which requires the robot to: (s1) navigate to a table, (s2) execute a pre-trained grasping policy to pick up an empty chip box, (s3) navigate to a trash bin, and (s4) execute a disposal policy to drop the chip box.
>
> **Experimental settings:**
> We evaluate this multi-stage task with 20 trials and under three methods: **Reachability, Human-intuition, and N2M**. The pose of the table, trash bin, and chip box are randomized in each trial. Detailed settings are provided in Appendix C.5.
>
> **Results:**
> The result is as follows. Note that due to limitations in the rebuttal, we cannot include figures and visual result analysis. We strongly encourage downloading our latest revision to view Section 5.3 Comprehensive Case 3.
>
> | **Method** | Half Task Success(s1+s2) | Full Task Success(s1+s2+s3+s4)  |
> |--------------|----------------------|-----------------------|
> | Reachability | 8/20 (40%) | 3/20 (15%) |
> | Human-intuition | 6/20 (30%) | 5/20 (25%) |
> | N2M | **18/20 (90%)** | **12/20 (60%)** |
>
> **Analysis:**
> The result reveals two critical insights:
>
> First, **sole reachability** proves insufficient for effective manipulation. **Even for humans** without prior knowledge of the policy, determining the optimal base pose is challenging. **Notably,** we observed that after several trials, humans begin to discern the policy's positional preferences, gradually achieving higher success rates. **This observation aligns with our design intuition** for N2M, which similarly learns these preferences through trials, or more formally, policy rollouts.
>
> Second, the **sequential nature of this multi-stage task** demonstrates how performance compounds across subtasks, highlighting the importance of optimizing base pose in each subtask. **Our proposed N2M module**, learning from policy rollouts, effectively captures the base preferences, substantially improving both subtask success rates and overall sequence completion. This underscores N2M's significance in multi-task mobile manipulation scenarios where sequential success is paramount.

---

### Official Review · Reviewer_tyZd · 2025-10-31

**Soundness:** 3
**Presentation:** 3
**Contribution:** 2
**Rating:** 4
**Confidence:** 4

**Summary:**

This paper identifies a critical misalignment in mobile manipulation pipelines, where navigation modules deliver the robot to a task area without considering the optimal initial pose for the subsequent manipulation policy. To bridge this gap, the authors introduce N2M, a transition module that re-positions the robot into a preferable initial pose after navigation. The core strength of N2M lies in its practical and efficient design, which operates on egocentric observations and demonstrates exceptional data efficiency and generalizability. Its key advantages are convincingly demonstrated through a dramatic performance improvement, elevating success rates from a baseline of 3% to 54% in one task, and showing reliable operation in unseen environments with only minimal data.

**Strengths:**

1. The field of mobile manipulation, which this paper addresses, represents a relatively nascent yet highly challenging research area.
2. The authors propose N2M, a transitional module designed to reposition the robot into a more favorable initial pose following the completion of navigation.
3. Extensive experimental results are provided to validate the effectiveness of the proposed method and the rationality of its design.

**Weaknesses:**

1. Does the pose in this paper refer to the position of the robot base or the position of the robot gripper?
2. The generalizability of the method to novel viewpoints should be further evaluated.
3. How would the system adjust the robot's position in case of partial occlusions?
4. Could the authors elaborate on the training methodology for the policy illustrated in Figure 5(b)?

**Questions:**

please see the weaknesses

**Details Of Ethics Concerns:**

no ethics concerns

---

> ### Author Response · Authors · 2025-11-23
> **Overview (1/n)**
>
> **Feedback from Reviewer 3**: Reviewer 3 noted that our research addresses a challenging problem in a nascent field, bridging the gap between navigation and manipulation policies. The reviewer understood the core ideas of the proposed method and acknowledged that the abundant simulation and physical experiments included in the paper effectively validated the proposed method from both qualitative and quantitative perspectives.
>
> **Concerns from Reviewer 3**: Reviewer 3 raised three questions about the paper's details: what specifically "robot pose" refers to, how to handle partial occlusion situations, and training details of the model in Fig.5(b). Additionally, the reviewer suggested further exploring the generalizability performance of the N2M method proposed in the paper.
>
> **Reply from Authors**: The authors provided detailed answers to the reviewer's questions and agreed that generalizability is an important feature of the proposed method. In fact, they have already conducted two sets of experiments demonstrating the remarkable generalizability of the proposed method from both qualitative (Fig.10 d-h) and quantitative (Fig.7) perspectives. They have also further investigated the reason for generalizability at the representation level (Fig.12). However, at the reviewer's request, they will add a new real-world experiment to further demonstrate N2M's generalizability performance.
>
> **Expectations from Authors**: The authors again thank Reviewer 3 for the time and effort spent improving this paper. However, they note that although Reviewer 3 understood the proposed method and acknowledged the paper's numerous supporting experiments, the contribution assessment (score 2) was not ideal. To address Reviewer 3's concerns, the authors summarized N2M's core contributions in their reply and used experimental results to demonstrate that the proposed method significantly outperforms all existing baselines. **The authors hope to receive more evaluation and suggestions from Reviewer 3 regarding the paper's contribution.** The authors have strong confidence in the value of their research problem, and their experimental results significantly outperform baselines, so they are particularly concerned about whether their work has been correctly delivered and is easily understood.

---

> ### Author Response · Authors · 2025-11-23
> **Reply to Strengths (2/n)**
>
> > The field of mobile manipulation, which this paper addresses, represents a relatively nascent yet highly challenging research area.
>
> Thank you for pointing this out. We especially tackle a critical yet highly underestimated problem in the domain of mobile manipulation of learning manipulation policy's preference. The proposed N2M method has two main features:
>
> 1. **Policy awareness**: Existing methods determine the robot pose without considering the manipulation policy's preference. This often results in poor policy performance, as visuomotor manipulation policies are highly sensitive to their initial pose.
> 2. **Practicality**: We provide and justify the critical challenges to making the solution easy to train and easy to deploy in the real world. Consequently, our proposed method shows 5 key features: broad applicability, real-time adaptability to environmental changes, viewpoint robustness, data efficiency, and generalizability.
>
>
> > The authors propose N2M, a transitional module designed to reposition the robot into a more favorable initial pose following the completion of navigation.
>
> We again highlight the importance of this problem. Due to the extreme sensitivity of learned manipulation policies, navigation often leads the robot to a very suboptimal position. With consideration of the policy's preference, it can drastically improve the performance (e.g., 3% to 54% in pick-and-place counter to cabinet task) as shown in Sections 4.2 and 5.1. We introduce unique challenges towards solving this problem and provide corresponding solutions.
>
> 1. **Adaptability to non-static environments**: The environment is typically non-static, therefore requiring the module to adapt to environmental changes. We tackle this by using ego-centric observations and generating predictions in a single forward pass.
> 2. **Multi-modality of preferable initial poses**: There may exist multiple preferable initial poses. To account for this, we predict the distribution of initial poses using GMM.
> 3. **Criterion of preferable initial poses**: Manipulation performance depends on multiple factors: policy architecture, training data distribution, robot configuration, task, and environment. Rather than attempting to model these complex relationships, we directly evaluate the pose through policy rollouts.
> 4. **Viewpoint robustness**: Since the robot navigation end poses can be anywhere within the task area, N2M needs to provide reliable predictions at various viewpoints. We introduce viewpoint augmentation to ensure robustness to viewpoint changes.
> 5. **Data efficiency**: Collecting rollouts requires substantial time and human effort, as each rollout must be monitored and manually labeled with success or failure. By simplifying the design to directly predict initial pose and with viewpoint augmentation, our method shows high data efficiency and remarkable generalizability.
>
>
> > Extensive experimental results are provided to validate the effectiveness of the proposed method and the rationality of its design.
>
> We appreciate the reviewer's acknowledgement of our experiments. In our paper, we first reanalyzed the challenges in base positioning and then addressed previous limitations by completely reparadigming the solution. The resulting method possesses several novel features that require thorough validation.
>
> To this end, we conducted 10 experiments across 9 tasks in both simulation and real-world settings, demonstrating that N2M exhibits all claimed features. The summary of features and their corresponding validation experiments is as follows:
>
> 1. **Broad Applicability**: Experiments in Section 4.2 and Section 5.1 show that our method is agnostic to policy architectures or tasks.
> 2. **Real-time Adaptation**: We demonstrate through Figure 15 and videos provided in our [anonymized website](https://nav2manip.github.io/) the real-time prediction and adaptation to dynamic environments.
> 3. **Viewpoint Robustness**: All experiments were conducted so that N2M gives a prediction from a widely randomized region spanning across the task area. We quantitatively show in Figures 5, 9, and also qualitatively show 10 consecutive successes in Figure 14 to show viewpoint robustness.
> 4. **Data Efficiency**: We provide detailed analysis for the data requirements in Figure 6 and also quantitatively analyze throughout all simulation (Section 4.2) and real-world experiments (Section 5.1, 5.2) that our method can give reliable predictions with only 10-20 data samples.
> 5. **Generalizability**: We quantitatively (Figure 7, 9) and qualitatively (Figure 10) analyze the generalizability of our proposed method, showing strong performance even in unseen environments.
>
> Collectively, these demonstrated features prove that our method is easy to train, straightforward to use, and therefore highly practical for real-world applications.

---

> ### Author Response · Authors · 2025-11-23
> **Reply to Weakness - 1 (3/n)**
>
> > Does the pose in this paper refer to the position of the robot base or the position of the robot gripper?
>
> The pose refers to the robot base pose in our work, as we aim to predict the manipulation policy's preference towards its base placement. Here are the specific settings we used.
>
> 1. **Simulation**: We consider SE(2) $(x,y,\theta)$ as the base pose
> 2. **Real-world**: We consider SE(2) + height $(x,y,\theta,z)$ as the base pose.
>
>
> > The generalizability of the method to novel viewpoints should be further evaluated.
>
> Thank you for your comment. We agree that the generalizability is an important feature of our method. In fact, we already included abundant experiments to demonstrate N2M's performance under various novel viewpoints and totally unseen environments:
>
> 1. **Unseen viewpoints**: Viewpoint robustness is demonstrated throughout all experiments (Sections 4.2, 4.3, 4.4, 5.1, and 5.2). As detailed in Appendix B.2 and C.2, the N2M predictions are generated from a highly randomized base pose, so each observation is unseen during inference. This shows that N2M provides consistent, accurate predictions regardless of the robot's placement, thereby confirming generalizability to observations under novel viewpoints.
> 2. **Unseen environments**: We also test the N2M module's generalizability in unseen environments, as shown in Sections 4.4, 5.1, and 5.2, both qualitatively and quantitatively. To further investigate the reason for N2M's generalizability, we computed and visualized the learned representations from the N2M encoder (Fig.12), which showed that the model specifically focuses on task-relevant key features in the task area.
>
> Although we have shown abundant experiments to demonstrate the generalizability of the N2M module, we highly appreciate the reviewer's comment, as we believe this is an important aspect of our proposed method. **We are currently conducting additional real-world experiments to further support our method and claim.** We also welcome more specific suggestions regarding the experiment design for better showing generalizability.

---

> ### Author Response · Authors · 2025-11-23
> **Reply to Weakness - 2 (4/n)**
>
> > How would the system adjust the robot's position in case of partial occlusions?
>
> We empirically observe that N2M demonstrates strong robustness to partial occlusions. We attribute this positive outcome to the following key reasons.
>
> 1. **No serious occlusion due to inference after navigation**: Our focus is the refinement of the robot's pose after navigation (as illustrated in Figure 1). We assume navigation has already guided the robot to a geometrically feasible position, as recent navigation algorithms [1-3] are capable of doing so. This setting ensures that no serious occlusion is present.
> 2. **Occlusion handled when training N2M**: By applying viewpoint augmentation across a large range of varying camera viewpoints, our training data naturally includes instances of partial occlusion. This process inherently trains N2M to be robust to partial occlusion in real-world scenarios.
>
> We acknowledge the reviewer's point to clearly understand the scope and limitations of our methodology. In response, we provide the following analysis of failure cases that empirically delineate the performance boundaries of our proposed method.
>
> 1. **Small objects**: Since we downsample the point cloud before providing it to the model, we observe that objects as small as a pen or an eraser are typically indistinguishable, leading to erroneous predictions.
> 2. **Far distance**: False predictions occur when the robot is positioned too far from the region of interest. This distance makes the region indistinguishable and, crucially, pushes the input outside the $2\text{m} \times 2\text{m}$ viewpoint sampling area used for augmentation, rendering it an out-of-distribution scenario.
> 3. **Noisy sensor**: Noisy estimates of the point cloud generally made the point cloud out of distribution, leading to noisy predictions. To resolve this, we used a high-performance Zed2 camera to observe high-quality point clouds.
> 4. **Manipulation policy limits**: This represents the most critical failure case observed, and it is highly relevant to the core motivation of our project. Even when our module provided a reliable prediction, we still observed manipulation failure due to the extreme sensitivity and limited capability of the underlying policy.
>
> [1] Openfmnav: Towards open-set zero-shot object navigation via vision-language foundation models
> [2] How to not train your dragon: Training-free embodied object goal navigation with semantic frontiers
> [3] Goat: Go to any thing
>
> > Could the authors elaborate on the training methodology for the policy illustrated in Figure 5(b)?
>
> Here are the details regarding both manipulation policy training and N2M module training, illustrated in Figure 5(b):
>
> 1. **Manipulation policy**: We train all the manipulation policies following the manual given in robocasa / robomimic. Using 3000 demonstrations generated from mimicgen, released by robocasa, we train each algorithm for 1000 epochs. We note that BC Transformer and ACT policy use absolute action, and diffusion policy uses relative action. [Click here to find official guidance of training manipulation policy](https://robocasa.ai/docs/use_cases/policy_learning.html)
> 2. **N2M module**: We train N2M based on the following order. (1) We rollout the trained manipulation policy with randomized base pose (Refer to Figure 13 for randomization criterion) and collect a few pairs of captured point clouds and successful initial poses (Refer to Section 3.3.1). (2) We sample M=300 viewpoints from the scene and augment the data as illustrated in Section 3.3.2 and Figure 4. (3) Finally,  train N2M network using our design explained in Section 3.2

---

> ### Author Response · Authors · 2025-11-23
> **Restating our contributions (5/n)**
>
> We noticed the reviewer gave a low score on contribution (score 2). To address reviewer's concern, we will restate and summarize our key contributions. We would like to strongly claim that our work is highly novel and offers strong advancements compared to previous methods.
>
> 1. **Identify Critical but Highly Underestimated Problem**: The misalignment between navigation and manipulation, stemming from the manipulation policy's extreme sensitivity to initial conditions, is a critical, yet currently overlooked problem. We thoroughly demonstrate that addressing this misalignment dramatically boosts performance, as shown in the comparison with the reachability baseline (e.g., Section 4.2 yields a performance increase from 3% to 54% in the *Pick-and-Place Counter to Cabinet* task).
> 2. **A Thorough Re-paradigm**: We systematically redesign the approach to the robot positioning problem, naturally overcoming limitations of previous methods. Moreover, we thoroughly investigate and list the technical challenges regarding this new problem, provide solutions for each of them, and provide meaningful insights towards our design choices (e.g., learning from rollouts, viewpoint augmentation).
> 3. **The Most Practical and Effective Solution**: We present a highly effective and practical solution that achieves notable improvements by directly addressing the fundamental limitations of prior works. Crucially, our N2M method demonstrates adaptability to dynamic environments, a unique advantage over all existing works. Evidence of this reliability is provided on our [**anonymized website (click here)**](https://nav2manip.github.io/), where each video showcases multiple consecutive successful predictions in dynamic settings.
>
> We believe the above explanation provides the necessary evidence to demonstrate the strong performance and contribution of our work.

---

> ### Author Response · Authors · 2025-11-23
> **Welcoming Additional Suggestions (n/n)**
>
> We are confident that **our work addresses a critical but highly underestimated problem** that existing methods fail to solve. We propose a promising and practical solution supported by ten comprehensive experiments designed to fully demonstrate its features, robustness, and practicality.
>
> We thank the reviewer for their time and thoughtful assessment. We would be extremely grateful for any further suggestions that could potentially enhance the clarity, rigor, or overall impact of our work. Your additional insights would be invaluable in significantly improving our work.
>
> 1. **Clarity and Interpretability**: We would greatly appreciate suggestions on any sections including the problem definition, method description, or results that hinder the paper's interpretability or overall flow.
> 2. **Experimental Support**: We seek the reviewer's judgment on whether all our claims are currently well-supported by the provided experimental results. If any specific claim lacks adequate empirical evidence, we are prepared to conduct additional analyses to strengthen our submission.

---

> ### Author Response · Authors · 2025-11-27
> **Result Update - Overview (1/4)**
>
> We provide additional experimental results regarding the following points:
>
> 1. **Demonstration of SOTA performance**: We compare our approach against the Reachability baseline (representing MoTo, MoManipVLA, and B*) and Mobi-$\pi$, the only policy-aware alternative. We evaluate both success rates and computational time to provide a comprehensive and fair comparison across methods. The results show that our method significantly outperforms these baselines.
> 2. **Justification of design choices**: We present detailed ablation studies that validate our design choices and provide comprehensive explanations for each component of our approach.
> 3. **Paper Revision**: We have incorporated the newly added **benchmark and ablation studies** into the paper, including detailed analysis and discussion. Additionally, we have addressed reviewer concerns about potentially **misleading claims**. We have also rewritten the **related work** section to clarify the categorization of previous methods and their distinctions from our approach. Furthermore, we have included several typical **failure cases** to provide readers with a more complete understanding of our method's limitations.
>
> Detailed revisions can be found in our **updated paper**, marked in blue.

---

> ### Author Response · Authors · 2025-11-27
> **Result Update - Benchmark and Analysis (2/4)**
>
> To further illustrate the distinctions between N2M and existing methods, we conducted benchmark experiments. We selected the CloseDrawer task, which is the **shared task** between Mobi-$\pi$ and our work, and **tested with two different policies**: BC Transformer (trained following RoboCasa's official guide and open-source data) and a diffusion policy (using Mobi-$\pi$'s open-source configuration and checkpoint). We performed **6 independent experiments**, each consisting of 50 inference trails, to calculate the **success rates** $S_{DP}$ and $S_{BC}$ for all methods under both policies. The experiments were conducted on an NVIDIA GeForce RTX 3090 24GB GPU and an AMD EPYC 7763 64-Core Processor. Benchmark results are recorded in the following table:
>
> | **Method**      | $S_\text{DP}$    | $S_\text{BC}$    | $T_\text{human}$ | $T_\text{train}$ | $T_\text{inference}$ |
> |-----------------|------------------|------------------|------------------|------------------|----------------------|
> | **Oracle**      | 0.93 ± 0.04      | 0.97 ± 0.02      | -                | -                | -                    |
> | **Reachability**| 0.16 ± 0.04      | 0.15 ± 0.03      | 0                | 0                | 0.72 ± 0.36 s      |
> | **Mobi-$\pi$**  | 0.24 ± 0.05      | 0.09 ± 0.03      | n×5 min          | n×7 min          | 273.52 ± 13.06 s           |
> | **N2M**         | **0.55 ± 0.07**  | **0.56 ± 0.04**  | 12.9 min         | 3.4 h            | **0.07 ± 0.03 s**  |
>
> **Success Rate Analysis**:
> **Reachability**-based method shows poor performance as it fails to consider the policy's pose preferences. Surprisingly, **Mobi-$\pi$** doesn't perform well under either policy **despite using its open-source code**. We attribute this to two reasons: first, observation similarity is just a heuristic of policy performance and cannot indicate the true policy performance; second, Mobi-$\pi$ uses a pretrained RGB encoder to extract RGB features, which might not be capable of distinguishing minor observation differences, as also discussed in their paper. In contrast, **N2M** learns pose preferences directly from actual policy rollouts, enabling more accurate prediction of preferred base positions and achieving state-of-the-art performance compared to existing methods.
>
> **Time Cost Analysis**:
> **Mobi-pi** requires prior scene reconstruction (5min scanning + 7min 3DGS training), and Bayesian optimization during inference (**273.52s for each inference**). Once the environment changes, users have to reconstruct the scene. We use $n$ to represent the number of changes. In contrast, **N2M** requires policy rollout collection (12.9min for 20 samples) and training (3.4 hours), but achieves rapid inference (**0.07s for each inference**), enabling real-time prediction in dynamic environments.
>
> In summary, **N2M accurately predicts high-success base pose for manipulation task with real-time inference**. In contrast, alternative approaches either suffer from low performance (reachability method) or slow inference speeds (Mobi-$\pi$). While N2M requires data collection, our proposed viewpoint augmentation technique substantially reduces the number of policy rollouts needed, making the approach more practical for real-world deployment.

---

> ### Author Response · Authors · 2025-11-27
> **Result Update - Ablation Study (3/4)**
>
> We further provide **ablations with naive augmentations, encoder choices, the use of pretrained weights, and the number of augmented viewpoints**. Detailed explanation and graph is also provided in our revised paper Section 6 and Appendix F.
>
> **Comparison with naive augmentations**
> | **Augmentation** | **Success Rate** |
> |------------------|------------------|
> | All              | **0.557 ± 0.035**|
> | w/o noise        | 0.537 ± 0.029    |
> | w/o pos          | 0.330 ± 0.038    |
> | w/o view         | 0.270 ± 0.040    |
>
> This result establishes viewpoint augmentation as essential for high performance. Although we compare different augmentation methods, we note that viewpoint augmentation is complementary with positional augmentation and random noise. The highest performance is attained when all augmentations are combined, demonstrating the synergy of these augmentations.
>
> **Point cloud encoder choices and pretrained weights**
> | **Encoder**         | **Success Rate** |
> |---------------------|------------------|
> | PointBERT           | **0.537 ± 0.029**|
> | PointNet++          | 0.467 ± 0.022    |
> | PointTransformerV3  | 0.120 ± 0.014    |
>
> | **Model**                  | **Success Rate** |
> |----------------------------|------------------|
> | N2M                        | **0.537 ± 0.029**|
> | N2M w/o pretrained PointBERT | 0.507 ± 0.032  |
>
> Although the PointBERT encoder with pretrained weights achieved the highest performance, the choice of encoder (PointBERT vs PointNet++) and use of pretrained weights did not critically affect the overall performance of N2M.
>
> **Number of augmented viewpoints**
> | **Viewpoints** | **Success Rate** |
> |----------------|------------------|
> | M=1            | 0.297 ± 0.029    |
> | M=10           | 0.397 ± 0.019    |
> | M=50           | 0.437 ± 0.044    |
> | M=100          | 0.467 ± 0.015    |
> | M=200          | 0.457 ± 0.022    |
> | M=300          | **0.537 ± 0.029**|
> | M=400          | **0.540 ± 0.035**|
> | M=500          | 0.483 ± 0.019    |
> | M=1000         | 0.493 ± 0.023    |
>
> This result shows that performance improves with a greater number of augmented views, saturating around $M=300$ views.

---

> ### Author Response · Authors · 2025-11-27
> **Result Update - Conclusion and Request for Reconsideration (4/4)**
>
> We thank the reviewer's constructive feedback. The newly added benchmark experiment and multiple ablation studies have substantially strengthened the clarity and principles underlying this paper. **We are confident that this work addresses a novel and important problem while providing a practical, innovative solution with state-of-the-art performance**. Our key contributions include:
>
> 1. **Identifying a critical yet underestimated problem**: We demonstrate that accounting for a manipulation policy's inherent preferences and sensitivities is crucial for successful mobile manipulation. Our experiments reveal that performance decreases dramatically without this consideration (as evidenced by the Reachability baseline), underscoring the necessity of our approach.
> 2. **Proposing a highly practical method with SOTA performance**: Our N2M method not only significantly outperforms geometry-based baselines but also surpasses the only existing policy-aware alternative, Mobi-$\pi$. The extensive benchmarks validate both the effectiveness and novelty of our policy-aware approach.
> 3. **Technical contribution**: We propose a novel viewpoint augmentation method that substantially improves viewpoint robustness and data efficiency while enhancing model generalization. Moreover, it can be seen as a general 3d data processing techniques, which has significant potential applications both within and beyond the domain discussed in the paper.
>
> We are confident that our findings and solutions will advance the field of mobile manipulation. All simulation and real-world implementations (including benchmark code) will be released upon paper acceptance. In light of these comprehensive enhancements and the demonstrated significance of our work, **we respectfully invite our reviewer to reconsider the paper's contributions and impact based on the revised experimental results and manuscript**.

---

> ### Author Response · Authors · 2025-12-02
> **Result update 2 - Further address the reviewer concern and paper updated**
>
> **One more new real-world experiment**
>
> We currently have a new real-world experiment (Section 5.3, Comprehensive Case 3) that we believe will further address the reviewer's concern about generalizability. As we introduced in Appendix C.5, the furniture, target object, and robot base are randomized within a 3m × 5m area for each trial, resulting in each inference of N2M occurring from an unseen viewpoint in a new scene.
>
> **Summarize existing experiments that show N2M's generalizability**
>
> We agree that the generalizability is an important feature of our method. In fact, we already included abundant experiments to demonstrate N2M's performance under various unseen viewpoints and totally unseen environments:
> 1. **Unseen viewpoints**: Viewpoint robustness is demonstrated throughout all experiments (Sections 4.2, 4.3, 4.4, 4.5, 5.1, 5.2, 5.3). As detailed in Appendix B.2 and C.2, the N2M predictions are generated from a highly randomized base pose, so each observation is unseen during inference. This shows that N2M provides consistent, accurate predictions regardless of the robot's placement, thereby confirming generalizability to observations under novel viewpoints.
> 2. **Unseen environments**: We also test the N2M module's generalizability in unseen environments, as shown in Sections 4.4, 5.1, 5.2, and 5.3, both qualitatively and quantitatively. To further investigate the reason for N2M's generalizability, we computed and visualized the learned representations from the N2M encoder (Fig.11), which showed that the model specifically focuses on task-relevant key features in the task area.
>
> **Discussion already updated in our latest revision**
>
> We already included the discussion (policy training explanation, occlusion, failure cases, new real-world experiment, and generalizability) in our paper. Please check our latest revision to get detailed information.

---

### Official Review · Reviewer_RoPP · 2025-11-01

**Soundness:** 3
**Presentation:** 3
**Contribution:** 2
**Rating:** 4
**Confidence:** 2

**Summary:**

In mobile manipulation, directly start manipulation from where the navigation ends may be ineffective, as the manipulation policy has preferred starting locations. To address this issue, this paper proposes to rollout the manipulation policy at different locations and fitting a distribution of successful locations, which could be used to generate navigation end points.

**Strengths:**

1. The paper is well written, with clear description of its technical challenges and solutions. All figures are helpful for getting across the proposed method and the experiment setups.
2. The authors conduct a large number of experiments to validate the proposed method, both in simulation and in the real world.

**Weaknesses:**

My major concern is the novelty of the proposed method. Though I am not very familiar with mobile manipultion field, according to my limited understanding, finding an appropriate starting pose for manipulation policy is a classic problems in the filed, and there are already several prior papers focusing on this challenge [1][2]. It's unclear to me, compared to them, in which way the proposed method is more novel and contributes to solving this questions.

[1] Iriondo, Ander, et al. "Learning positioning policies for mobile manipulation operations with deep reinforcement learning." International journal of machine learning and cybernetics 14.9 (2023): 3003-3023.
[2] Jauhri, Snehal, Jan Peters, and Georgia Chalvatzaki. "Robot learning of mobile manipulation with reachability behavior priors." IEEE Robotics and Automation Letters 7.3 (2022): 8399-8406.

**Questions:**

See weakness.

---

> ### Author Response · Authors · 2025-11-23
> **Overview (1/n)**
>
> **Feedback from Reviewer 2**: After examining our paper, Reviewer 2 found the research problem clearly defined and appreciated the paper's good structure that first lists the five technical challenges in the problem and then provides corresponding solutions. Additionally, Reviewer 2 believed that the paper demonstrated, through extensive experiments and illustrations, that the proposed method effectively addresses the problems and challenges presented.
>
> **Concerns from Reviewer 2**: However, Reviewer 2 considered robot base positioning a classic topic that has already been extensively studied, with mature solutions available. Therefore, the reviewer found it difficult to assess the significance of revisiting this problem and distinguishing our approach from previous methods. Additionally, Reviewer 2 questioned the innovativeness of our method and cited two relevant publications, requesting clarification on how our approach differs from them.
>
> **Reply from Authors**: The authors sincerely thank Reviewer 2 for their valuable time and clearly expressed concerns. They agree that "finding an appropriate starting pose is a classic problem" but explain why revisiting this problem is necessary. They then demonstrate how their paper completely redesigns the solution, pioneering a new perspective that naturally overcomes previous limitations. Finally, they clearly differentiate their approach from the two papers cited by Reviewer 2.
>
> **Expectations from Authors**: The authors again thank Reviewer 2 for the time and effort spent reviewing this paper. They would like to know if their response has addressed Reviewer 2's concerns. Additionally, they look forward to receiving more suggestions from Reviewer 2, particularly regarding any gaps in content delivery. The authors are highly confident in the value of their research problem and their experimental results significantly outperform baselines, so they are particularly concerned about whether their work has been delivered correctly and is easily understood.

---

> ### Author Response · Authors · 2025-11-23
> **Reply to Strength (2/n)**
>
> > The paper is well written, with clear description of its technical challenges and solutions.
>
> We thank the reviewer for pointing this out. We summarize the technical challenges of designing a module to bridge the critical misalignment between navigation and manipulation, and the corresponding design choices we took to solve each of the challenges.
>
> 1. **Adaptability to non-static environments**: The environment is typically non-static, therefore requiring the module to adapt to environmental changes. We tackle this by using ego-centric observations and generating predictions in a single forward pass.
> 2. **Multi-modality of preferable initial poses**: There may exist multiple preferable initial poses. To account for this, we predict the distribution of initial poses using GMM.
> 3. **Criterion of preferable initial poses**: Manipulation performance depends on multiple factors: policy architecture, training data distribution, robot configuration, task, and environment. Rather than attempting to model these complex relationships, we directly evaluate the pose through policy rollouts.
> 4. **Viewpoint robustness**: Since the robot navigation end poses can be anywhere within the task area, N2M needs to provide reliable predictions at various viewpoints. We introduce viewpoint augmentation to ensure robustness to viewpoint changes.
> 5. **Data efficiency**: Collecting rollouts requires substantial time and human effort, as each rollout must be monitored and manually labelled with success or failure. By simplifying the design to directly predict initial pose and viewpoint augmentation, our method shows high data efficiency and remarkable generalizability.
>
> > All figures are helpful for getting across the proposed method and the experiment setups. The authors conduct a large number of experiments to validate the proposed method, both in simulation and in the real world.
>
> We appreciate the reviewer's acknowledgement of our figures and experiments. In our paper, we first reanalyzed the challenges in base positioning and then addressed previous limitations by completely re-paradigming the solution. The resulting method possesses several novel features that require thorough validation.
>
> To this end, we conducted 10 experiments across 9 tasks in both simulation and real-world settings, demonstrating that N2M exhibits all claimed features. The summary of features and their corresponding validation experiments is as follows:
>
> 1. **Broad Applicability**: Experiments in Section 4.2 and Section 5.1 show that our method is agnostic to policy architectures or tasks.
> 2. **Real-time Adaptation**: We demonstrate through Figure 15 and videos provided in our [anonymized website](https://nav2manip.github.io/) the real-time prediction and adaptation to dynamic environments.
> 3. **Viewpoint Robustness**: All experiments were conducted so that N2M gives prediction from a widely randomized region spanning across the task area. We quantitatively show in Figure 5, 9 and also qualitatively show 10 consecutive successes in Figure 14 to show viewpoint robustness.
> 4. **Data Efficiency**: We provide detailed analysis for the data requirements in Figure 6 and also quantitatively analyze throughout all simulation (Section 4.2) and real-world experiments(Section 5.1 5.2) that our method can give reliable predictions with only 10-20 data samples.
> 5. **Generalizability**: We quantitatively (Figure 7, 9) and qualitatively (Figure 10) analyze the generalizability of our proposed method, showing strong performance even in unseen environments.
>
> Collectively, these demonstrated features prove that our method is easy to train, straightforward to use, and therefore highly practical for real-world applications.

---

> ### Author Response · Authors · 2025-11-23
> **Reply to Weakness (3/n)**
>
> > My major concern is the novelty of the proposed method. Though I am not very familiar with mobile manipultion field, according to my limited understanding, finding an appropriate starting pose for manipulation policy is a classic problems in the filed,
>
> We agree that "finding an appropriate starting pose" is a classic problem. However, we revisited this problem because recent changes in manipulation policies have made previous methods ineffective:
> 1. **Evolution of policies**: Traditional manipulation relied on predicting grasping poses and using model-based planners for collision-free trajectories. These planner-based methods were limited to simple pick-and-place tasks. Recently, data-driven visuomotor policies (BC[1-2], VLA[3-4], RL-finetuned[5-6]) have become mainstream, which are more capable but have different positioning requirements.
> 2. **Ineffectiveness of previous methods**: Prior approaches using reachability calculations were sufficient for planner-based policies but fail with modern visuomotor manipulation policies, which are highly sensitive to starting pose due to their training data distributions. By accounting for this difference, our method demonstrated a drastic performance increase, specifically from 3% to 54% in the Pick-and-Place Counter to Cabinet task compared to the reachability baseline representing prior geometric approaches.
> 3. More details about previous approaches can be found in Section 2.3 of our paper.
>
> Our work makes three primary contributions:
>
> 1. **Problem Identification**: We identify and analyze the critical misalignment between navigation and manipulation modules. With experiments, we show that this misalignment stems from the manipulation policy's extreme sensitivity to its initial pose, a factor largely overlooked by prior work.
> 2. **A Thorough Re-paradigm**: We systematically redesign the approach to the robot positioning problem, naturally overcoming limitations of previous methods:
>    - **Getting rid of assumptions**: By learning from policy rollouts rather than comparing observation similarity, N2M maintains policy awareness while eliminating dependency on policy training data.
>    - **Efficient Inference**: By employing direct distribution prediction instead of searching+iterative optimization, N2M achieves significantly faster inference and lower computational cost.
>    - **Data Efficiency Innovation**: By proposing viewpoint-based data augmentation, N2M enhances both data efficiency and viewpoint robustness, substantially reducing data collection burden for preference learning. As Reviewer 1 noted, our proposed data augmentation method has broader value beyond the problem we are tackling.
> 3. **Comprehensive Validation**: We implement our method using a simple architecture (encoder+MLP+GMM) and conduct extensive experiments demonstrating N2M's new features: applicability across different robot configurations, tasks, and policy types; remarkable data efficiency and generalizability; viewpoint robustness; and real-time inference based on ego-centric observations.
>
>
> [1] Learning fine-grained bimanual manipulation with low-cost hardware
> [2] Diffusion policy: Visuomotor policy learning via action diffusion
> [3] $\pi_{0.5}$: a Vision-Language-Action Model with Open-World Generalization
> [4] Gemini robotics: Bringing ai into the physical world
> [5] $\pi^{*} _ {0.6}$: a VLA That Learns From Experience
> [6] RL-100: Performant Robotic Manipulation with Real-World Reinforcement Learning
>
> > And there are already several prior papers focusing on this challenge [1][2]. It's unclear to me, compared to them, in which way the proposed method is more novel and contributes to solving these questions.
>
> Prior works, including the two papers cited by the reviewer, are ineffective when applied to modern visuomotor manipulation policies.
>
> 1. **Failing to account for modern visuomotor policies**: The work [1] uses Inverse-Reachability-Map to determine robot's placement and [2] uses reachability criterion as a reward to train positioning policy. Both works are one of the representative reachability-based methods that aren't applicable to modern visuomotor policy due to high sensitivity towards starting pose.
> 2. **Data efficiency and practicality**: The work in [3], like [2], uses RL to train a transition policy, leveraging the learned policy's signal as a reward. However, the well-known sample inefficiency of RL severely limits their practicality in real-world deployment. In contrast, our method addresses this by simplifying the problem to direct target pose prediction and incorporating viewpoint augmentation, making it highly data-efficient and practical for real-world use, as detailed in the Introduction and Sections 4.3 and 6.
>
> [1] Robot learning of mobile manipulation with reachability behavior priors.
> [2] Learning positioning policies for mobile manipulation operations with deep reinforcement learning.
> [3] Composing complex skills by learning transition policies

---

> ### Author Response · Authors · 2025-11-23
> **Welcoming Additional Suggestions (n/n)**
>
> **We are confident** that our work addresses a critical but highly underestimated problem that **ALL** of the existing methods fail to solve. We propose a promising and practical solution, supported by 10 comprehensive experiments that fully demonstrate its features, robustness, and practicality.
>
> However, **our primary concern** now is whether our paper effectively delivers our ideas to readers. We would be highly grateful for any further suggestions that could potentially enhance the clarity, rigor, or overall impact of our work. Your additional insights would be invaluable in significantly improving our work.
>
> 1. **Clarity and Interpretability**: We would greatly appreciate suggestions on any sections, including the introduction, related work, problem definition, method description, or results that hinder the paper's interpretability or overall flow.
> 2. **Experimental Support**: We seek the reviewer's judgment on whether all our claims are currently well-supported by the provided experimental results. If any specific claim lacks adequate empirical evidence, we are prepared to conduct additional analyses to strengthen our submission.

---

> > ### Comment · Reviewer_RoPP · 2025-11-26
> >
> > I appreciate the detailed explanation from the authors. My concerns are resolved and I have raised the score accordingly.

---

> > > ### Author Response · Authors · 2025-11-27
> > > **Additional Results and Appreciation for Reconsideration**
> > >
> > > We are deeply grateful for your time in reassessing our work and are delighted that our contributions have been recognized with an improved evaluation.
> > >
> > > We have added more experiments and ablation studies and uploaded the latest revision of our paper. Here is a summary of the new results:
> > > 1. **Demonstration of SOTA performance**: We compare our approach against the Reachability baseline (representing MoTo, MoManipVLA, and B*) and Mobi-$\pi$, the only policy-aware alternative. We evaluate both success rates and computational time to provide a comprehensive and fair comparison across methods. The results show that our method significantly outperforms these baselines.
> > > 2. **Justification of design choices**: We present detailed ablation studies that validate our design choices and provide comprehensive explanations for each component of our approach.
> > > 3. **Paper Revision**: We have incorporated the newly added **benchmark and ablation studies** into the paper, including detailed analysis and discussion. Additionally, we have addressed reviewer concerns about potentially **misleading claims**. We have also rewritten the **related work** section to clarify the categorization of previous methods and their distinctions from our approach. Furthermore, we have included several typical **failure cases** to provide readers with a more complete understanding of our method's limitations.
> > >
> > > All revisions are highlighted in blue in our updated manuscript for your convenience.
> > >
> > > We sincerely appreciate your continued engagement with our work and thank you for acknowledging its value to the field.

---

### Official Review · Reviewer_77AG · 2025-11-01

**Soundness:** 3
**Presentation:** 3
**Contribution:** 3
**Rating:** 4
**Confidence:** 3

**Summary:**

This paper addresses the misalignment between navigation and manipulation in mobile robots, where navigation focuses solely on reaching task areas without considering which initial poses are preferable for executing manipulation policies. The authors propose N2M (Navigation-to-Manipulation), a transition module that predicts multi-modal distributions of preferable initial poses using Gaussian Mixture Models from ego-centric RGB point clouds. The key technical contributions include: (1) learning pose preferences directly from policy rollouts, treating policies as black boxes without requiring access to internals or training data, (2) a viewpoint augmentation strategy that renders observations from multiple angles during training, enabling viewpoint robustness and data efficiency with only 10-15 rollouts, and (3) real-time adaptation through single forward pass prediction from ego-centric observations without global scene reconstruction. Experimental validation includes simulation results across 4 tasks and 3 policy architectures in RoboCasa showing dramatic improvements (3% → 54% success rate on PnPCounterToCab), and real-world deployment on 5 tasks demonstrating 10 consecutive successes with randomized configurations and generalization to unseen environments.

**Strengths:**

**Originality**

Novel problem formulation: The paper clearly identifies and formalizes the navigation-manipulation misalignment problem, which has been overlooked despite its criticality. The key innovation is **learning pose preferences directly from rollouts**, treating policies as black boxes—a creative departure from prior work requiring specific policy types (RL for value functions, IL for distributional similarity) or access to training data.

Technical innovation: The combination of GMM-based distribution modeling with viewpoint augmentation is elegant. Notably, viewpoint augmentation simultaneously addresses multiple challenges (robustness, data efficiency, generalization)—an emergent insight that represents a valuable contribution beyond the primary problem.

**Quality**

Comprehensive experiments spanning 4 simulation tasks with 3 policy architectures, systematic data efficiency analysis, controlled generalization studies (texture vs. layout), and 5 real-world tasks. The ablation study confirms viewpoint augmentation's importance, and attention visualizations (Figure 12) provide interpretability. The paper also acknowledges N2M exceeding oracle performance and discusses policy preferences versus training distributions. However, real-world tasks (b)-(e) use manual rules rather than actual policies, slightly weakening end-to-end claims. More failure case analysis would strengthen the work.

**Clarity**

Well-structured: Clear progression from problem to solution to validation. The five-challenge framework effectively organizes the approach, and visualizations strongly support claims—particularly the multi-modality illustration (Figure 2), success heatmaps (Figure 9), and generalization demonstrations (Figure 10).

Minor gaps: Some technical details need clarification—coordinate frame transformations, kernel number selection rationale (K=2 vs K=1), and the quantitative impact of augmentation ratio M=300 on final performance.

**Significance**

Practical impact: Addresses a fundamental bottleneck in mobile manipulation with dramatic improvements (3% → 54% success rates). The data efficiency (10-15 rollouts), broad applicability across tasks/policies/hardware, and ego-centric design lower adoption barriers significantly.

Research contributions: Opens research directions on viewpoint augmentation's effectiveness and rollout-based preference learning. The finding that policy preferences diverge from training distributions has broader implications for understanding learned behaviors.

Scope limitations: While valuable, the work improves existing modular pipelines rather than proposing transformative paradigms. End-to-end approaches may eventually supersede such bridging solutions. Real-world experiments remain in controlled lab settings.

Overall, this is a solid, practical contribution that will benefit the community but may not be groundbreaking.

**Weaknesses:**

**Problem Importance is Questionable**

Base placement is already solved: Methods like B* [1] provide efficient and optimal base placement. The dramatic improvement over the "reachability baseline" (3% → 54%) likely reflects a poorly implemented baseline rather than fundamental limitations of geometric approaches. The paper must compare against state-of-the-art optimization methods (B*, modern inverse reachability maps) to justify that learning is necessary.

Ego-centric observation is not an advantage: This is presented as a benefit but is actually a limitation: (1) Modern robots have mapping capabilities, (2) Methods like Mobi-π plan optimal docking from anywhere using scene reconstruction, while N2M requires navigating close first, (3) No evidence that ego-centric is faster or more robust than global methods.

**Low Technical Novelty**

Standard components: Point-BERT (pre-trained) + MLP + GMM prediction. GMM for robotic poses is established (Bahl et al. 2023), and learning from binary rollout labels is basic preference learning. Architecture (Figure 3) contains no innovation.

"Learning from rollouts" overclaimed: This is standard reward-free imitation from success/failure labels. Predicting poses instead of actions (vs. Lee et al. 2019) is a simplification, not innovation.

Viewpoint augmentation under-analyzed: The only interesting aspect but: (1) No ablation on M=300, (2) No comparison with simpler augmentations, (3) No mechanistic explanation, (4) "10-15 rollouts" is misleading—actually 3000-4500 samples after M=300× augmentation.

**Insufficient Empirical Validation**

Missing critical baselines: No comparison with B* [1], Mobi-π, MoTo, or validation of Lee et al.'s "1000+ rollouts" claim.

Weak real-world validation: Only 1 of 5 tasks uses actual learned policy—tasks (b)-(e) use manual rules. Generalization tested on only 5 similar scenes (Section 4.4) or qualitative only (Section 5.2).

Misleading data efficiency: Claims "10-15 rollouts" but ignores M=300 multiplier. No human time comparison or empirical validation against alternatives.

No statistical rigor: Success rates lack error bars or significance tests despite high variance (Figure 6).


**Missing Ablations**

No ablations on: (1) Encoder choice (Point-BERT vs. alternatives), (2) Loss regularization terms and hyperparameters, (3) GMM kernel number K, (4) Fine-tuning vs. frozen encoder.

**Questions:**

**Questions**

**Q1: Quantitative Comparison with Optimization Methods**
How does N2M compare with state-of-the-art optimization methods like B* [1]? The 3% → 54% improvement over your reachability baseline suggests a poorly implemented baseline rather than fundamental limitations of geometric approaches. Please provide: (a) success rates for B* or modern inverse reachability methods on your tasks, (b) computational cost comparison, (c) specific scenarios where learning demonstrably outperforms optimization, or (d) acknowledgment if the reachability baseline doesn't represent state-of-the-art geometric methods.

**Q2: Viewpoint Augmentation Analysis and Data Requirements**
Why is M=300 necessary, and what are the true data requirements? Claiming "10-15 rollouts" when M=300 produces 3000-4500 training samples is misleading. Please provide: (a) ablation varying M ∈ {10, 50, 100, 300} showing performance vs. data trade-offs, (b) comparison with simpler augmentations (random jittering, noise), (c) total human time for data collection vs. alternatives, (d) mechanistic explanation of why geometric rendering improves generalization beyond just data quantity.

**Q3: Real-World Validation with Learned Policies**
Why do only 1 of 5 real-world tasks use actual learned manipulation policies? Tasks (b)-(e) use manual rules, weakening end-to-end claims. Please provide: (a) quantitative success rates with learned policies for at least 2 additional tasks, (b) validation that manual rules accurately approximate policy preferences, or (c) explicit acknowledgment as a limitation with discussion of why full policy validation wasn't feasible.

**Q4: Missing Comparisons and Ablations**
What are the quantitative comparisons with related work and ablations on design choices? Please provide: (a) success rate and efficiency comparison with Mobi-π and MoTo on shared tasks, (b) encoder comparison (Point-BERT vs. PointNet++/PointTransformer), (c) loss function regularization ablation with hyperparameter values (αw, αdist, αmode), (d) empirical validation of data efficiency claims vs. Lee et al. 2019, or acknowledge which comparisons/ablations are infeasible.

**Suggestions**

**S1: Provide Statistical Rigor and Failure Analysis**
All success rates lack confidence intervals despite high variance. Include: (a) error bars and significance tests for all results, (b) failure mode breakdown (why 0.0 success in some Figure 9 cells?), (c) conditions where viewpoint robustness or generalization fails, (d) sensitivity to depth noise and partial occlusions. This analysis is critical for understanding practical limitations and deployment reliability.

---

> ### Author Response · Authors · 2025-11-22
> **Overview (1/n)**
>
> **Feedback from Reviewer 1**: Reviewer 1 carefully examined our paper, fully understood our method, acknowledged our innovation and experimental results, and provided numerous specific and valid improvement suggestions.
>
> **Concerns from Reviewer 1**: However, Reviewer 1 questioned the value of the problem itself, suggesting that the robot base positioning problem has already been solved. Additionally, the low success rate of the reachability baseline raised concerns that it might be due to poor implementation by the authors. Regarding comparisons with previous methods, Reviewer 1 felt the baselines were too limited and recommended adding quantitative comparisons with more methods, such as Mobi-π.
>
> **Reply from Authors**: The authors sincerely appreciate Reviewer 1's rigour and attentiveness, and accept most of the suggestions provided. Regarding the importance of the research problem, the authors noted that Reviewer 1 may have underestimated its significance and overestimated the reliability of existing methods. The authors have clarified the importance and long-term value of the research problem and explained the implementation approach for the reachability baseline.
>
> **Next Step of Authors**: Furthermore, the authors will promptly provide comprehensive benchmark results, ablation studies, and supplementary real-world experiments.
>
> **Expectations from Authors**: The authors again thank Reviewer 1 for the time and effort spent reviewing this paper. **They would like to know if their response has addressed Reviewer 1's concerns.** The authors are highly confident in the value of their research problem, and their experimental results significantly outperform baselines, so they are particularly concerned about whether their work has been delivered correctly and is easily understood.

---

> ### Author Response · Authors · 2025-11-22
> **Reply to Strength - Originality - 1 (2/n)**
>
> > Novel problem formulation: The paper clearly identifies and formalizes the navigation-manipulation misalignment problem, which has been overlooked despite its criticality. The key innovation is learning pose preferences directly from rollouts, treating policies as black boxes—a creative departure from prior work requiring specific policy types (RL for value functions, IL for distributional similarity) or access to training data.
>
> Thank you for your clarification. Our work tackles two main limitations of prior work.
>
> 1. **Lack of Policy Awareness**: Most existing methods determine the robot pose without considering the manipulation policy's preference. This often results in poor policy performance, as visuomotor manipulation policies are highly sensitive to the initial pose where they are executed.
> 2. **Impracticality of Policy-Aware Alternative**: The only concurrent policy-aware method, Mobi-π, relies on Gaussian Splats-based scene reconstruction and assumes the access to training datasets, which limits Mobi-π's applicability to only IL-based policies trained with small-scale data and static scenes.
>
> In contrast, our work makes three primary contributions:
>
> 1. **Problem Identification**: We identify and analyze the critical misalignment between navigation and manipulation modules. With experiments, we show that this misalignment stems from the manipulation policy's extreme sensitivity to its initial pose, a factor largely overlooked by prior work.
> 2. **A Thorough Re-paradigm**: We systematically redesign the approach to robot positioning problem, naturally overcoming limitations of previous methods:
>    - **Getting rid of assumptions**: By learning from policy rollouts rather than comparing observation similarity, N2M maintains policy awareness while eliminating dependency on policy training data.
>    - **Efficient Inference**: By employing direct distribution prediction instead of searching+iterative optimization, N2M achieves significantly faster inference and lower computational cost.
>    - **Data Efficiency Innovation**: By proposing viewpoint-based data augmentation, N2M enhances both data efficiency and viewpoint robustness, substantially reducing data collection burden for preference learning.
> 3. **Comprehensive Validation**: We implement our method using a simple architecture (encoder+MLP) and conduct extensive experiments demonstrating N2M's new features: applicability across different robot configurations, tasks, and policy types; remarkable data efficiency and generalizability. Additionally, by abandoning batch search and iterative optimization approaches, our method can predict based on real-time scenes using only ego-centric views.
>
> Specifically, our method shows clear strengths over Mobi‑π:
>
> 1. **Adaptiveness to non-static environment**: N2M predicts the policy‑preferred pose in a single forward pass from an egocentric point cloud, eliminating costly scene reconstruction. In contrast, Mobi‑π must rescan and recompute COMAP and rendered fields whenever the scene changes — a 5×10 m scene typically takes 10-30 minutes per reconstruction — making it impractical for non-static environments.
> 2. **No Assumption on Manipulation Policy**: N2M treats the manipulation policy as a black box and does not assume availability of its training data. This makes N2M applicable across diverse policy types — including model‑based, imitation‑ and reinforcement‑learning-based, and modern Vision‑Language‑Action (VLA) policies, whether zero‑shot or RL‑fine‑tuned. For example, **π0.5**[1] and **Gemini Robotics**[2] rely on large‑scale web and cross-embodiment data pretraining, and **π*0.6**[3] improves via online experience from reinforcement learning.
>
> [1] π0.5: a Vision-Language-Action Model with Open-World Generalization
> [2] Gemini robotics: Bringing ai into the physical world
> [3] π*0.6: a VLA That Learns From Experience

---

> ### Author Response · Authors · 2025-11-22
> **Reply to Strength - Originality - 2 (3/n)**
>
> > Technical innovation: The combination of GMM-based distribution modeling with viewpoint augmentation is elegant. Notably, viewpoint augmentation simultaneously addresses multiple challenges (robustness, data efficiency, generalization)—an emergent insight that represents a valuable contribution beyond the primary problem.
>
>
> Thank you for acknowledging the technical novelty of our approach. We're pleased to share our discovery process:
>
> 1. **Initial problem identification**: After proposing to learn policy preferences from rollout results, we realized that the robot's navigation end pose could be anywhere within the task area. Collecting rollout results from all possible viewpoints would significantly increase the data collection burden.
> 2. **Technical solution**: To address this challenge, we designed a method to render observations from different viewpoints within the task area. We are particularly fortunate because the labels (robot poses) remain consistent across rendered viewpoints—the scene itself determines the policy pose preference rather than the observation viewpoint.
> 3. **Surprising benefits**: During implementation, we discovered that viewpoint augmentation not only improved viewpoint robustness and data efficiency but also enhanced model generalization. This makes intuitive sense because our rendering preserves the core interaction area while background elements naturally vary with different viewpoints. This encourages the model to identify truly important aspects of the observation.
> 4. **Experimental validation**: We conducted numerous experiments to demonstrate this generalizability: In our real-world Toybox Handover task, we collected 15 successful rollout samples in one room (Fig.10(d)) and trained our N2M model with this limited dataset. The same trained N2M model could then function well in a zero-shot manner across four completely different, unseen environments (Fig.10(e-h)). We also conducted quantitative experiments in simulation to estimate N2M's generalizability (Fig. 7). To further investigate this phenomenon, we computed and visualized the learned representations from the N2M encoder (Fig.12), which showed that the model specifically focuses on task-relevant key features in the task area.

---

> ### Author Response · Authors · 2025-11-22
> **Reply to Strength - Quality (4/n)**
>
> > Comprehensive experiments spanning 4 simulation tasks with 3 policy architectures, systematic data efficiency analysis, controlled generalization studies (texture vs. layout), and 5 real-world tasks. The ablation study confirms viewpoint augmentation's importance, and attention visualizations (Figure 12) provide interpretability.
>
> We appreciate the reviewer's feedback on our experiments. To clarify, we have structured our experiments to demonstrate specific features of our method, and we summarize the purpose of each experiment below.
>
> 1. **Broad Applicability**: Experiments in Section 4.2 and Section 5.1 show that our method is agnostic to policy architectures or tasks.
> 2. **Viewpoint Robustness**: All experiments were conducted so that N2M gives prediction from a widely randomized region spanning across task area. We quantitatively show in Figure 5, 9 and also qualitatively show 10 consecutive successes in Figure 14 to show viewpoint robustness.
> 3. **Real-time Adaptation**: We demonstrate through Figure 15 and videos provided in our [anonymized website](https://nav2manip.github.io/) the real time prediction and adaptation to dynamic environments.
> 4. **Data Efficiency**: We provide detailed analysis for the data requirements in Figure 6 and also quantitatively analyze throughout all simulation (Section 4.2) and real world experiments(Section 5.1 5.2) that our method can give reliable predictions with very small amount of data.
> 5. **Generalizability**: We quantitatively (Figure 7, 9) and qualitatively (Figure 10) analyze the generalizability of our proposed method showing strong performance even in unseen environments.
>
> All of these features demonstrate that our method is easy to train and use, contributing to its practicality.
>
> > The paper also acknowledges N2M exceeding oracle performance and discusses policy preferences versus training distributions.
>
> Thank you for acknowledging our experimental results that shows the necessity of learning from rollouts. We would like to clarify two major benefits of learning from rollouts
>
> 1. **Removal of policy assumptions**: This design choice allows us to make no assumptions on the policy's internal design. This is especially beneficial when deploying models like pretrained VLAs or RL-based policies, where training data is often unavailable or not used. In contrast, our closest work, Mobi-pi, is not compatible with these types of policy designs because of its requirement on training data.
> 2. **Provides accurate signals for preference**: As the reviewer mentioned, our findings demonstrate that rollouts are a direct and effective signal for the policy's preference. This is validated when our method's performance surpasses that of the oracle baseline (Section 4.2), suggesting rollouts capture the policy's true behavioral preferences more accurately than the oracle itself.
>
>
> > However, real-world tasks (b)-(e) use manual rules rather than actual policies, slightly weakening end-to-end claims. More failure case analysis would strengthen the work.
>
> As noted in the paper, the purpose of real-world tasks (b)-(e) is to qualitatively analyze our method's performance. Our detailed quantitative analysis is provided separately through simulation (Section 4.2) and our primary real-world task (Section 5.1).
>
> However, we understand the reviewer's concerns regarding more extensive real-world validation and will provide additional results as soon as possible.

---

> ### Author Response · Authors · 2025-11-22
> **Reply to Strength - Clarity (5/n)**
>
> > Minor gaps: Some technical details need clarification—coordinate frame transformations, kernel number selection rationale (K=2 vs K=1), and the quantitative impact of augmentation ratio M=300 on final performance.
>
> Thank you for pointing this out. We will include the following explanations in our paper.
>
> 1. **Ego-centric coordination**: Our method utilizes the robot's ego-centric (base) frame of reference for both the input point cloud and the output pose. This ensures the model's predictions are invariant to the robot's global position. Therefore, the transformation process follows: (1) capture point cloud, (2) transform its coordinate to robot base coordinate, (3) N2M module processes this input and generates a pose prediction, which is also relative to the robot's base, (4) this predicted pose is then transformed back into the global coordinate system, and (5) the robot adjusts its global pose to match this target.
> 2. **Empirical selection of kernel number ($K$)**: The number of kernels ($K$) is set empirically based on human observation. We set $K=2$ for the CloseDrawer task, as we manually identified two significant modes (left and right). All other tasks exhibited only one dominant mode, so we used $K=1$. Users can similarly determine $K$ by observing the number of significant modes during their own rollout collection.
> 3. **Quantitative Impact of Augmentation Ratio ($M$)**: We are conducting an experiment on the impact of the augmentation ratio $M$, evaluating $M \in \{10, 50, 100, 200, 300, 400, 500, 1000\}$ on the 'Pick-and-place Counter to Cabinet' simulation task. We will provide the results as soon as possible.
>
>
> >Practical impact: Addresses a fundamental bottleneck in mobile manipulation with dramatic improvements (3% → 54% success rates). The data efficiency (10-15 rollouts), broad applicability across tasks/policies/hardware, and ego-centric design lower adoption barriers significantly.
>
> Thanks for acknowledging the experimental results in the paper. By doing simulation and real-world experiments, the authors want to show two things:
> 1. **Demonstrating the Necessity of Learning Policy Preference**: As mentioned in the comments, we show the limitations of geometry-based methods. We demonstrate the necessity of learning the manipulation policy's preference by showing the dramatic performance difference between approaches that do and do not consider it.
> 2. **Practicality**: We demonstrate that viewpoint augmentation yields significant data efficiency. This simplifies the training process, thereby lowering the barrier for applying N2M to new tasks, policies, and environments and ensuring its practicality.
>
> >Research contributions: Opens research directions on viewpoint augmentation's effectiveness and rollout-based preference learning. The finding that policy preferences diverge from training distributions has broader implications for understanding learned behaviours.
>
> We appreciate the reviewer's recognition of this point. We wish to emphasize the research contributions and technical insights that our work provides to the mobile manipulation community.
>
> 1. **Identification of a critical yet underestimated problem**: We identify a crucial but highly underestimated problem of learning the manipulation policy's preference. We show that popular geometry-based methods are insufficient for determining an optimal pose for executing the manipulation policy because they fail to account for the policy's preferences, thereby underscoring the importance of this issue.
> 2. **Providing key technical insights**: As mentioned in the comment, we show how viewpoint augmentation improves the data efficiency of our design and provide insights demonstrating that learning from rollouts yields more accurate predictions.

---

> ### Author Response · Authors · 2025-11-22
> **Reply to Strength - Significance (6/n)**
>
> >Scope limitations 1: While valuable, the work improves existing modular pipelines rather than proposing transformative paradigms. End-to-end approaches may eventually supersede such bridging solutions.
>
> We understand the reviewer's concern that our work addresses the integration issue in modular mobile manipulation systems rather than developing an end-to-end solution that might eventually supersede modular approaches. We offer two perspectives in response:
>
> **Current reality of mobile manipulation**
>
> While end-to-end approaches have proven effective for many tasks, their advantages remain unverified for mobile manipulation systems. To our knowledge, ALL current mobile manipulation systems still employ modular designs, for several reasons:
> 1. **Existence of high-performance modules**: Mobile manipulation systems comprise complex, functionally independent modules (localization/mapping, task planning, navigation, and manipulation), each with relatively mature and reliable solutions already.
> 2. **Fundamental difference in task**: Navigation and manipulation tasks differ fundamentally—the former focuses on collision-free path planning to target locations, while the latter emphasizes environmental interaction and contact. They also prioritize different aspects of environmental representation. Moreover, there's a severe shortage of training data that simultaneously covers both navigation and manipulation to support end-to-end model training.
> 3. **Enabling efficient and easy integration**: Current navigation modules can reliably guide robots to task areas but struggle to connect to downstream manipulation policy execution effectively. Our work identifies this specific gap and proposes a practical solution that significantly improves task success rates in mainstream mobile manipulation systems, offering substantial practical value.
> 4. **Both modularization and end-to-end are valuable**: This doesn't mean we oppose end-to-end approaches. From the current technological standpoint, we believe the motivation and value of end-to-end solutions for navigation and manipulation remain to be demonstrated. Both approaches deserve encouragement and attention without bias. In unexplored domains, we should avoid presupposing future technical solutions to discover more possibilities.
> 5. **Clarifying misconceptions**: We believe the reviewer may be referring to recent VLA models (such as π0.5) when discussing existing end-to-end methods. However, we contend that these models are more accurately categorized as (whole-body) manipulation policies, as they are not designed for, nor capable of, long and complex navigation tasks. Therefore, to the best of our knowledge, no existing system truly integrates both long-range navigation and manipulation end-to-end.
>
> **Enduring value of our work**
>
> Even if end-to-end approaches represent the future of mobile manipulation systems, our work maintains significant value:
>
> 1. **Problem identification**: We clearly identify and experimentally verify the severity of the navigation-manipulation integration problem in modular systems, an issue previously underestimated and lacking practical solutions.
> 2. **Framework for understanding**: We clearly define five challenges in this problem domain, helping the research community better understand the issue and potentially develop improved solutions based on our foundation.
> 3. **Broader technical contribution**: We discover that viewpoint augmentation can enhance sample efficiency, viewpoint robustness, and generalization capabilities, a technical contribution extending beyond our specific problem domain.
>
>
> > Scope limitations 2: Real-world experiments remain in controlled lab settings. Overall, this is a solid, practical contribution that will benefit the community but may not be groundbreaking.
>
> Thank you for acknowledging the practical value of our work. We are committed to providing additional experimental results to demonstrate the reliability of our N2M module further. Regarding the assessment that our work may not be groundbreaking, we offer two points that may help reconsider the long-term significance of our contribution:
> 1. **Current state of the Mobile Manipulation**: As noted in our previous response, we have highlighted the challenges facing end-to-end approaches in mobile manipulation system design. To our knowledge, **ALL** existing mobile manipulation systems employ modular designs. The problem we identify and address represents an underestimated yet critical issue in current modular mobile manipulation systems.
> 2. **Comparative advantage**: The reviewer may overestimate the effectiveness and practicality of existing methods in this domain. We will conduct detailed benchmark experiments to demonstrate that N2M's efficiency and effectiveness substantially exceed those of previous methods. In addressing this specific problem, we believe N2M represents a groundbreaking advancement.

---

> ### Author Response · Authors · 2025-11-22
> **Reply to Weakness - 1 (7/n)**
>
> > Base placement is already solved: Methods like B* [1] provide efficient and optimal base placement.
>
> We appreciate the reviewer's comment. However, we strongly refute the statement that the base placement problem is solved and claim that it remains a critical, open challenge.
>
> 1. **Huge sensitivity of learned policies**: Though geometry-based methods can be a suitable choice for planner-based policies, learned policies show strong sensitivity towards base placement, as shown in our experiments, making geometry-based base placement a very suboptimal choice. We show in Section 4.2, 5.1 that with the consideration of policy's preference, we can boost the performance significantly.
> 2. **Amplification of sensitivity due to mobility**: Learned policies, even in the domain of table-top manipulation, are known to be highly sensitive to environment configurations, including camera viewpoint, object position, color, and lighting. This fundamental sensitivity is dramatically amplified in mobile manipulation, as the robot's mobility makes maintaining any consistency in environment configurations impossible. This heightened sensitivity represents a critical challenge that must be accounted for to determine base placement, a crucial factor entirely overlooked by conventional geometry-based methods.
>
>
> > The dramatic improvement over the "reachability baseline" (3% → 54%) likely reflects a poorly implemented baseline rather than fundamental limitations of geometric approaches.
>
> We thank the reviewer for their rigorousness. However, we assert that this **dramatic improvement strongly reflects the fundamental limitations of geometric approaches**. To address potential concerns regarding the comparison, we clarify the methodology used to determine the base position of the reachability baseline (details are provided in Appendix B.2 and C.2). While the explanation focuses on the simulation experiment, we confirm that the exact same procedure was conducted for the real-world setup as well.
>
> 1. **Defining the Reachable Randomization Region**: In our RoboCasa simulation, which uses the Franka Panda arm (reach: $0.855 \text{m}$) mounted on an Omron mobile base, we define the randomization region for the robot's base placement as a $1 \text{m} \times 1 \text{m}$ square centered at the target object or furniture. This region is chosen to guarantee reachability. The maximum distance from the center of the region to any corner (the half-diagonal) is calculated as $0.5\sqrt{2} \text{m} \approx 0.707 \text{m}$, which is safely within the $0.855 \text{m}$ reach of the Franka arm.
> 2. **Defining Angular Variance for Visibility**: Based on the camera intrinsics used, the calculated horizontal angle of view is approximately $115.63^{\circ}$. We perturb the robot's orientation by an angular variance of $\pm 15^{\circ}$ (a $30^{\circ}$ total span) relative to the original base placement. Given the wide horizontal FOV, this restricted perturbation range ensures the target object or furniture remains visible within the camera's viewpoint.
> 3. **Collision Filtering**: Infeasible base poses are filtered by verifying potential collisions in the occupancy grid map.
>
> We therefore assert that the significant performance gain observed in our method is real and is not attributable to poor baseline implementation. Instead, the gain is a direct consequence of the manipulation policy's extreme sensitivity to its initial base placement. We, however, understand and respect the reviewer's concern and will provide a comparison with the baseline calculated with rigorous Inverse kinematics.
>
> > The paper must compare against state-of-the-art optimization methods (B*, modern inverse reachability maps) to justify that learning is necessary.
>
> Although we claim that our original reachability baseline remains valid, we will provide additional baselines below to support this claim.
>
> 1. **IK-based Reachability Baseline**: To establish a more rigorous comparison, we will provide a new baseline based on the Inverse Kinematic (IK) calculation and collision check. This approach will represent prior works—such as MoTo, MoManipVLA, and IRM that rely solely on geometry-based methods to determine the robot's base position.
> 2. **Evaluation Against Mobi-pi Baselines**: We will provide additional baselines, including the only concurrent policy-aware method, Mobi-pi, to quantitatively show the superior performance of our method.
>
> We will fill in the chart below for comparison.
>
> | Method | N2M | Mobi-pi | Reachability (IK-based) |
> |---|---|---|---|
> | Policy Awareness | O | O | X |
> | Success Rate | - | - | - |
> | Preparing Time | - | - | -|
> | Inference Time | - | - | -|

---

> ### Author Response · Authors · 2025-11-22
> **Reply to Weakness - 2 (8/n)**
>
> > Ego-centric observation is not an advantage: This is presented as a benefit, but is actually a limitation: (1) Modern robots have mapping capabilities, (2) Methods like Mobi-π plan optimal docking from anywhere using scene reconstruction, while N2M requires navigating close first, (3) No evidence that ego-centric is faster or more robust than global methods.
>
> We appreciate the reviewer's concern regarding the use of ego-centric observation. However, we note that there appears to be a significant misunderstanding regarding the differences between Mobi-π and N2M. Below, we first clarify their relationship and then address the reviewer's specific concerns.
>
> 1. **Same goal towards policy-aware base placement**: Mobi-π and N2M are concurrent works, both aiming to improve manipulation policy success rates by selecting appropriate policy-aware base positions. Their fundamental difference lies in how they determine suitable robot base positions, leading to substantially different outcomes.
> 2. **Assumptions on manipulation policy**: Mobi-$\pi$ determines a suitable robot position by comparing current observations with training data similarity. This requires access to policy training data, restricting Mobi-π to small-scale imitation learning policies, which is incompatible with state-of-the-art manipulation policies that are often pre-trained on vast internet data and demonstrate zero-shot capabilities (e.g., π₀.₅[1], Gemini Robotics[2]) or are refined through continuous experiential feedback (e.g., π₀.₆[3], RL-100[4]). In contrast, N2M makes no assumptions about robot hardware, tasks, or policy types, making it applicable to cutting-edge manipulation policies.
> 3. **Time consumption during inference**: Similarity-based method in Mobi-$\pi$ leads to significantly large inference time. Because they calculate scores for a given position rather than directly predicting a desired pose, they require an extensive search and an iterative sampling method to determine the position with the highest score. This also requires reconstructing the entire scene, which is highly time-consuming. Mobi-π must first scan the entire environment and reconstruct it using 3DGS. This restricts Mobi-π to static environments, requiring complete rescanning whenever the environment changes. On the other hand, N2M directly predicts the distribution of suitable base poses from ego-centric observations.
> 4. **Requirement on training**: The only weakness of N2M is that it requires rollout collection and training unlike Mobi-$\pi$. However, we actively reduce data requirements by viewpoint augmentation. N2M requires only 10-20 successful rollout samples to deliver reliable predictions.
>
> Regarding above relationship between Mobi-$\pi$ and N2M, we provide explanations regarding reviewer's concern on our design choice of using ego-centric observations.
>
> 1. **Modern robot's mapping capability**: The map required by Mobi-$\pi$ (a 3D Gaussian Splatting (3DGS) map) is fundamentally incompatible with standard navigation maps, making it insufficient for immediate use. First, the map typically used during navigation is not a 3DGS representation; it is often a 2D occupancy grid or a sparse point cloud. Next, achieving the high-quality 3DGS required by Mobi-$\pi$ demands a sufficient quantity of images captured from diverse viewpoints. This necessitates an extensive active exploration phase during navigation. This active data collection for 3DGS places a non-trivial extra burden on the navigation system. Therefore, simply using the map obtained during standard navigation does not solve the problem of map creation for Mobi-$\pi$.
> 2. **Requirement for reaching the task area**: First of all, this is not a limitation as current navigation methods[5-7] are already capable of doing so. Moreover, our work is compatible with the global map, as we can sample from it.
> 3. **Speed comparison**: The time difference between ego-centric and global map-based interaction position prediction is not significant in itself. Rather, the scanning and construction of high-precision 3DGS maps is the fundamental factor preventing Mobi-π from operating in non-static environments. We emphasize ego-centric observation because N2M can make predictions based on real-time scene changes, not because global optimization is computationally slower than ego-centric processing.
>
> We believe this clarification regarding the reviewer's concerns is valuable and will be incorporated into the paper to help readers quickly understand the essential differences between Mobi-π and N2M. We again thank the reviewer for the thoughtful and in-depth questions.
>
> [4] RL-100: Performant Robotic Manipulation with Real-World Reinforcement Learning
> [5] Openfmnav: Towards open-set zero-shot object navigation via vision-language foundation models
> [6] How to not train your dragon: Training-free embodied object goal navigation with semantic frontiers
> [7] Goat: Go to any thing

---

> ### Author Response · Authors · 2025-11-22
> **Reply to Weakness - 3 (9/n)**
>
> > Standard components: Point-BERT (pre-trained) + MLP + GMM prediction. GMM for robotic poses is established (Bahl et al. 2023), and learning from binary rollout labels is basic preference learning. Architecture (Figure 3) contains no innovation.
>
> We acknowledge that the architectural design itself is not particularly novel. However, we argue that the architecture is not our primary contribution. As the reviewer noted, our main contribution lies in opening a critical new research problem and providing a highly practical and effective solution to it.
>
> 1. **Problem Identification**: We identify and analyze the critical misalignment between navigation and manipulation modules. We demonstrate that this misalignment stems from the manipulation policy's extreme sensitivity to its initial pose, a factor largely ignored by previous methods.
> 2. **A Practical Solution (N2M)**: We propose N2M, a novel method that directly addresses this problem. N2M stands out as a highly practical solution, demonstrating broad applicability, remarkable data efficiency, and robust generalization. It achieves real-time performance and maintains robustness across different viewpoints.
> 3. **Viewpoint augmentation**: We propose a novel method to improve data efficiency and robustness. This is our main technical contribution, which can be potentially applied to various other domains.
>
>
> > "Learning from rollouts" overclaimed: This is standard reward-free imitation from success/failure labels. Predicting poses instead of actions (vs. Lee et al. 2019) is a simplification, not innovation.
>
> We understand and admit the reviewer's concern. Although it might be a simple idea, our contribution lies in empirically proving their benefit and practical effectiveness in solving this critical problem.
>
> 1. **Agnosticity to task and policy**: While optimization-based methods are primarily suited for planner-based policies (or yield suboptimal results, as shown by the reachability baseline in Sections 4.2 and 5.1), and the only policy aware method, Mobi-pi, is restricted to policies trained via imitation learning, our approach is robustly applicable across a wide spectrum of tasks and policy designs.
> 2. **Simplification yielding extreme data efficiency**: The synergy between this design simplification and our proposed viewpoint augmentation technique results in extreme data efficiency in learning, which we empirically prove in our experiments (Section 4.2, 4.3, 5.1, 5.2, 6).
>
> We argue that learning from rollout is a crucial design choice for practicality.
>
>
> > Viewpoint augmentation under-analyzed: The only interesting aspect but: (1) No ablation on M=300, (2) No comparison with simpler augmentations, (3) No mechanistic explanation,
>
> Thank you for pointing this out. We will provide fthe ollowing additional experiments
>
> 1. **Ablation on viewpoint number**: We will provide additional results showing the effect of $M$. We will conduct experiments on $M \in \{10, 50, 100, 200, 300, 500, 1000\}$
> 2. **Comparison with simpler augmentations**: We will provide a comparison with additional augmentation methods, such as adding noise.
> 3. **Mechanistic explanation**: The main intuition behind the success of viewpoint augmentation towards generalization is that it enforces the model to focus on important aspects of the scene. By generating multiple perspectives of the same underlying context, the technique **forces the network to filter out irrelevant features and focus only on the task-relevant features of the scene**, leading to robust prediction regardless of the robot's position and enhancing generalization. This result is also analyzed in Section 6 as we prove that the model is learned to focus on the important region of the scene.
>
> Moreover, we reclaim our contribution beyond mere viewpoint augmentation.
>
> 1. **Problem Identification**: We assert that this problem identification itself is highly novel. The extreme sensitivity of manipulation policies is a critical factor in the mobile manipulation domain that has been broadly overlooked. By successfully identifying and focusing on this issue, we believe our work paves the way for future research in this essential direction.
> 2. **Detailed proof and analysis of our design choice**: We first show the core challenges inherent to solving this problem. We then rigorously justify every key design choice, demonstrating its contribution to overcoming these challenges and ensuring the method's overall practicality. This systematic approach confirms that our work provides a highly practical solution to this fundamental problem.
>
> > (4) "10-15 rollouts" is misleading—actually 3000-4500 samples after M=300× augmentation.
>
> We understand the reviewer's concern and will clarify the distinction in the paper. However, our primary emphasis remains on **how much data human should collect**, as it critically determines the practicality of the method.

---

> ### Author Response · Authors · 2025-11-22
> **Reply to Weakness - 4 (10/n)**
>
> > Missing critical baselines: No comparison with B*, Mobi-π, MoTo
>
> Thank you for your recommendation. The paper we provided focused on presenting the main features of our proposed method. We admit a lack of comparison with prior works and will provide quantitative results of baselines.
>
> 1. **Success rate**: We will compare the success rate, showing how well each method estimates policy preference
> 2. **Inference time & human time**: We will provide the time required to generate predictions for fair comparison and justify the practicality of our method
> 3. **IK-based Reachability Baseline**: This approach will represent prior works—such as MoTo, MoManipVLA, and B* that rely solely on geometry-based methods to determine the robot's base position.
>
> > Weak real-world validation: Only 1 of 5 tasks uses the actual learned policy—tasks (b)-(e) use manual rules. Generalization tested on only 5 similar scenes (Section 4.4) or qualitative only (Section 5.2).
>
> We would like to list all the experiments we have conducted that support all the features and advantages of N2M.
>
> 1. **(Simulation) Broad Applicability**: We quantitatively analyze N2M on diverse tasks and policies and prove that our method has broad applicability (Section 4.2)
> 2. **(Simulation) Data Efficiency**: We vary the number of rollouts and plot the success rate to estimate the number of rollouts required to train the N2M module (Section 4.3)
> 3. **(Simulation) Generalizability**: We provide a quantitative analysis on varying environment layouts and textures, validating the generalizability of our proposed method. (Section 4.4)
> 4. **(Real-world) Real-world validation**: We quantitatively assess the performance of N2M on the "Lamp Retrieval" task, validating its performance in the real world. (Section 5.1)
> 5. **(Real-world) Generalization (3 variations)**: We quantitatively analyze the generalizability of N2M, the capability of identifying preferable initial poses in unseen cells. (Section 5.1)
> 6. **(Real-world) Qualitative analysis**: We qualitatively analyze data efficiency and generalizability of our proposed method in the real world. (Section 5.2)
> 7. **(Real-world) Real-time prediction**: We evaluate N2M in a dynamically changing environment to show real-time adaptation of its prediction (Appendix E)
> 8. **(Real-world) Viewpoint robustness**: We vary the initial pose for every prediction generated to rigorously demonstrate the viewpoint robustness of our method. This procedure confirms that our predictions remain consistent regardless of the initial viewpoint. (All experiments)
>
> We understand and respect the reviewer's concern and will try our best to provide more real-world validations. However, we will prioritise providing baselines and an ablation study.
>
> > Misleading data efficiency: Claims "10-15 rollouts" but ignores M=300 multiplier. No human time comparison included.
>
> We argue this is not misleading as our main emphasis is on how much data human should collect. However, we will clarify the difference in our paper. We will further provide a comparison of human time and inference time between N2M and Mobi-pi as follows.
>
>
> > No statistical rigor: Success rates lack error bars or significance tests despite high variance (Figure 6).
>
> Thank you for pointing this out. We will update all the graphs with error bars!
>
> > No ablations on: (1) Encoder choice (Point-BERT vs. alternatives), (2) Loss regularization terms and hyperparameters, (3) GMM kernel number K, (4) Fine-tuning vs. frozen encoder.
>
> Thank you for the suggestion. We will provide additional ablation on the components of our method.

---

> ### Author Response · Authors · 2025-11-22
> **Reply to Questions - 1 (11/n)**
>
> > Q1(a): Quantitative Comparison with Optimization Methods How does N2M compare with state-of-the-art optimization methods like B*. The 3% → 54% improvement over your reachability baseline suggests a poorly implemented baseline rather than fundamental limitations of geometric approaches. Please provide: (a) success rates for B* or modern inverse reachability methods on your tasks,
>
> As mentioned above, we provide detailed criteria regarding the reachability baseline and claim that it does represent current geometry and optimization-based methods. **Please refer to our reply on weakness for detailed analysis on why our criterion guarantees reachability.** However, we will provide the IK-based reachability baseline to strengthen our claim.
>
> > Q1(b) computational cost comparison,
>
> We provide a comparison between alternatives as mentioned in the table above.
>
> > Q1\(c) specific scenarios where learning demonstrably outperforms optimization, or
>
> We maintain a robust claim that our approach yields superior performance over traditional optimization methods in **all but the planner-based policy scenario**. Crucially, planner-based policies exhibit severe limitations in applicability, necessitating the use of learning-based manipulation policies for broader deployment. The inherent sensitivity of learning-based policies requires accurate preference determination, thereby necessitating the use of our method in the most relevant use cases.
>
> > Q1(d) acknowledgment if the reachability baseline doesn't represent state-of-the-art geometric methods.
>
> We will implement the IK-based reachability baseline to represent the state-of-the-art geometric methods [8-10].
>
> [8] Rm4d: A combined reachability and inverse reachability map for common 6-/7-axis robot arms by dimensionality reduction to 4d
> [9] MoTo: A Zero-shot Plug-in Interaction-aware Navigation for General Mobile Manipulation
> [10] Momanipvla: Transferring vision-language-action models for general mobile manipulation
>
> > Q2(a) Viewpoint Augmentation Analysis and Data Requirements Why is M=300 necessary, and what are the true data requirements? Claiming "10-15 rollouts" when M=300 produces 3000-4500 training samples is misleading. Please provide: (a) ablation varying M ∈ {10, 50, 100, 300} showing performance vs. data trade-offs,
>
> To understand the impact of our augmentation strategy, we will perform an ablation study varying $M$, the number of augmented viewpoints, using the set $M \in \{10, 50, 100, 200, 300, 400, 500, 1000\}$.
>
>
> > Q2(b) comparison with simpler augmentations (random jittering, noise),
>
> We first want to clarify that these simpler augmentations are **not competing methods but are complementary to our viewpoint augmentation** that can be utilized in parallel. We appreciate the reviewer's suggestion and will provide additional results demonstrating the positive synergy of incorporating simple augmentations such as random noise and positional augmentation alongside our viewpoint augmentation.
>
>
> > Q2\(c) total human time for data collection vs. alternatives,
>
> As mentioned above, we will also provide a comparison between alternatives.
>
>
> > Q2(d) mechanistic explanation of why geometric rendering improves generalization beyond just data quantity.
>
> The key motivation behind the success of viewpoint augmentation is as follows.
>
> **Enforcing the model to focus on task-relevant features**: By generating multiple perspectives of the same underlying context, the technique forces the network to filter out irrelevant features and focus only on the task-relevant features of the scene, leading to robust prediction. It is also analyzed in Section 6 as we prove that the model is learned to focus on the important region of the scene.

---

> ### Author Response · Authors · 2025-11-22
> **Reply to Questions - 2 (12/n)**
>
> > Q3(a): Real-World Validation with Learned Policies Why do only 1 of 5 real-world tasks use actual learned manipulation policies? Tasks (b)-(e) use manual rules, weakening end-to-end claims. Please provide: (a) quantitative success rates with learned policies for at least 2 additional tasks,
>
> As previously explained, we have abundant experiments to support and prove all our claims both in simulation and the real world. However, we appreciate and understand the reviewer's point and will try our best to provide more real-world validations. Though we will prioritize providing baselines and an ablation study.
>
>
> > Q3(b) validation that manual rules accurately approximate policy preferences, or Q3\(c) explicit acknowledgment as a limitation with discussion of why full policy validation wasn't feasible.
>
> The main focus of real-world experiments (b)-(e) is to qualitatively assess the performance and generalizability of our proposed method. Therefore, we qualitatively evaluate the predictions generated from diverse scenes and provide our manual criterion for doing so.
>
> Again, we appreciate and understand the reviewer's concern and will try our best to provide more real-world validations, but will prioritise providing baselines and an ablation study.
>
>
> > Q4(a): Missing Comparisons and Ablations What are the quantitative comparisons with related work and ablations on design choices? Please provide: (a) success rate and efficiency comparison with Mobi-π and MoTo on shared tasks, Q4(b) encoder comparison (Point-BERT vs. PointNet++/PointTransformer), Q4\(c) loss function regularization ablation with hyperparameter values (αw, αdist, αmode), Q4(d) empirical validation of data efficiency claims vs. Lee et al. 2019, or acknowledge which comparisons/ablations are infeasible.
>
> Thank you for the suggestion. We will provide the following ablations.
>
> 1. **Baselines**: We will conduct experiments to compare with alternative methods such as Mobi-pi and geometric methods
> 2. **Encoder comparison**: We will conduct experiments with additional point cloud encoders, such as PointNet++ and PointTransformer, and compare the performance.
> 3. **Number of viewpoint augmentations**: We will ablate the number of viewpoints used to augment and provide justification for specific value selection $M=300$
> 4. **Simpler augmentation methods**: We will show the effect of simpler alternative augmentations, together with viewpoint augmentation.
> 5. **Use of pretrained PointBERT weight**: We will ablate the effect of the pretrained weight for the point cloud encoder.
>
> Although we will try our best to provide ablations for other design choices as well, we acknowledge that we will focus on the above-noted experiments, as they address more important aspects of our method.

---

> ### Author Response · Authors · 2025-11-22
> **Reply to Suggestions (n/n)**
>
> > S1(a): Provide Statistical Rigor and Failure Analysis All success rates lack confidence intervals despite high variance. Include: (a) error bars and significance tests for all results,
>
> Thank you for the suggestion. We will recalculate the statistics of our results and include error bars in the paper!
>
> > S1(b) failure mode breakdown (why 0.0 success in some Figure 9 cells?),
>
> There were two main failure modes in Figure 9.
>
> 1. **Extreme sensitivity of manipulation policy**: We use the top-left cell as a prime example of this failure mode. Despite N2M generating predictions that were qualitatively well-aligned with the policy's desired outcome, the policy itself failed to execute. This discrepancy underscores the extreme sensitivity of the manipulation policy.
> 2. **False prediction though geometrically feasible**: For cells not observed during training, the N2M module occasionally produced false predictions that did not align with the policy's observed preference. Critically, we note that these false predictions were often qualitatively hard to distinguish from successful ones without prior knowledge of the manipulation policy's specific learned biases.
>
> These failure modes again highlight the importance of our problem. We would like to emphasize the following point from this analysis.
>
> - **Current manipulation policies are also very sensitive**: We utilised the $\pi_0$[11] policy, the SOTA manipulation policy when we submitted this paper, which was finetuned with 600 demonstrations we manually collected (50 per cell), the model exhibited extreme sensitivity. This outcome strongly highlights the critical importance of policy awareness for accurately determining initial base poses.
>
> > S1\(c) conditions where viewpoint robustness or generalization fails, (d) sensitivity to depth noise and partial occlusions. This analysis is critical for understanding practical limitations and deployment reliability.
>
> We provide an explanation regarding factors that may affect the performance of N2M.
>
> 1. **Small objects**: Since we downsample the point cloud before providing it to the model, we observe that objects as small as a pen or an eraser are typically indistinguishable, leading to erroneous predictions.
> 2. **Far distance**: False predictions occur when the robot is positioned too far from the region of interest. This distance causes the region to become indistinguishable and, crucially, pushes the input outside the $2\text{m} \times 2\text{m}$ viewpoint sampling area used for augmentation, making it an out-of-distribution scenario.
> 3. **Noisy sensor**: Noisy estimates of the point cloud generally made the point cloud out of distribution, leading to noisy predictions. To resolve this, we used a high-performance Zed2 camera to observe high-quality point clouds.
> 4. **Manipulation policy limits**: This represents the most critical failure case observed, and it is highly relevant to the core motivation of our project. Even when our module provided a reliable prediction, we still observed manipulation failure due to the extreme sensitivity and limited capability of the underlying policy.
>
> [11] Black, Kevin, et al. "$\pi_0$: A Vision-Language-Action Flow Model for General Robot Control." arXiv preprint arXiv:2410.24164 (2024).

---

> ### Author Response · Authors · 2025-11-27
> **Result Update - Overview (1/4)**
>
> We provide additional experimental results regarding the following points:
>
> 1. **Demonstration of SOTA performance**: We compare our approach against the Reachability baseline (representing MoTo, MoManipVLA, and B*) and Mobi-$\pi$, the only policy-aware alternative. We evaluate both success rates and computational time to provide a comprehensive and fair comparison across methods. The results show that our method significantly outperforms these baselines.
> 2. **Justification of design choices**: We present detailed ablation studies that validate our design choices and provide comprehensive explanations for each component of our approach.
> 3. **Paper Revision**: We have incorporated the newly added **benchmark and ablation studies** into the paper, including detailed analysis and discussion. Additionally, we have addressed reviewer concerns about potentially **misleading claims**. We have also rewritten the **related work** section to clarify the categorization of previous methods and their distinctions from our approach. Furthermore, we have included several typical **failure cases** to provide readers with a more complete understanding of our method's limitations.
>
> Detailed revisions can be found in our **updated paper**, marked in blue.

---

> ### Author Response · Authors · 2025-11-27
> **Result Update - Benchmark and Analysis (2/4)**
>
> To further illustrate the distinctions between N2M and existing methods, we conducted benchmark experiments. We selected the CloseDrawer task, which is the **shared task** between Mobi-$\pi$ and our work, and **tested with two different policies**: BC Transformer (trained following RoboCasa's official guide and open-source data) and a diffusion policy (using Mobi-$\pi$'s open-source configuration and checkpoint). We performed **6 independent experiments**, each consisting of 50 inference trails, to calculate the **success rates** $S_{DP}$ and $S_{BC}$ for all methods under both policies. The experiments were conducted on an NVIDIA GeForce RTX 3090 24GB GPU and an AMD EPYC 7763 64-Core Processor. Benchmark results are recorded in the following table:
>
> | **Method**      | $S_\text{DP}$    | $S_\text{BC}$    | $T_\text{human}$ | $T_\text{train}$ | $T_\text{inference}$ |
> |-----------------|------------------|------------------|------------------|------------------|----------------------|
> | **Oracle**      | 0.93 ± 0.04      | 0.97 ± 0.02      | -                | -                | -                    |
> | **Reachability**| 0.16 ± 0.04      | 0.15 ± 0.03      | 0                | 0                | 0.72 ± 0.36 s      |
> | **Mobi-$\pi$**  | 0.24 ± 0.05      | 0.09 ± 0.03      | n×5 min          | n×7 min          | 273.52 ± 13.06 s           |
> | **N2M**         | **0.55 ± 0.07**  | **0.56 ± 0.04**  | 12.9 min         | 3.4 h            | **0.07 ± 0.03 s**  |
>
> **Success Rate Analysis**:
> **Reachability**-based method shows poor performance as it fails to consider the policy's pose preferences. Surprisingly, **Mobi-$\pi$** doesn't perform well under either policy **despite using its open-source code**. We attribute this to two reasons: first, observation similarity is just a heuristic of policy performance and cannot indicate the true policy performance; second, Mobi-$\pi$ uses a pretrained RGB encoder to extract RGB features, which might not be capable of distinguishing minor observation differences, as also discussed in their paper. In contrast, **N2M** learns pose preferences directly from actual policy rollouts, enabling more accurate prediction of preferred base positions and achieving state-of-the-art performance compared to existing methods.
>
> **Time Cost Analysis**:
> **Mobi-pi** requires prior scene reconstruction (5min scanning + 7min 3DGS training), and Bayesian optimization during inference (**273.52s for each inference**). Once the environment changes, users have to reconstruct the scene. We use $n$ to represent the number of changes. In contrast, **N2M** requires policy rollout collection (12.9min for 20 samples) and training (3.4 hours), but achieves rapid inference (**0.07s for each inference**), enabling real-time prediction in dynamic environments.
>
> In summary, **N2M accurately predicts high-success base pose for manipulation task with real-time inference**. In contrast, alternative approaches either suffer from low performance (reachability method) or slow inference speeds (Mobi-$\pi$). While N2M requires data collection, our proposed viewpoint augmentation technique substantially reduces the number of policy rollouts needed, making the approach more practical for real-world deployment.

---

> ### Author Response · Authors · 2025-11-27
> **Result Update - Ablation Study (3/4)**
>
> We further provide **ablations with naive augmentations, encoder choices, the use of pretrained weights, and the number of augmented viewpoints**. Detailed explanation and graph is also provided in our revised paper Section 6 and Appendix F.
>
> **Comparison with naive augmentations**
> | **Augmentation** | **Success Rate** |
> |------------------|------------------|
> | All              | **0.557 ± 0.035**|
> | w/o noise        | 0.537 ± 0.029    |
> | w/o pos          | 0.330 ± 0.038    |
> | w/o view         | 0.270 ± 0.040    |
>
> This result establishes viewpoint augmentation as essential for high performance. Although we compare different augmentation methods, we note that viewpoint augmentation is complementary with positional augmentation and random noise. The highest performance is attained when all augmentations are combined, demonstrating the synergy of these augmentations.
>
> **Point cloud encoder choices and pretrained weights**
> | **Encoder**         | **Success Rate** |
> |---------------------|------------------|
> | PointBERT           | **0.537 ± 0.029**|
> | PointNet++          | 0.467 ± 0.022    |
> | PointTransformerV3  | 0.120 ± 0.014    |
>
> | **Model**                  | **Success Rate** |
> |----------------------------|------------------|
> | N2M                        | **0.537 ± 0.029**|
> | N2M w/o pretrained PointBERT | 0.507 ± 0.032  |
>
> Although the PointBERT encoder with pretrained weights achieved the highest performance, the choice of encoder (PointBERT vs PointNet++) and use of pretrained weights did not critically affect the overall performance of N2M.
>
> **Number of augmented viewpoints**
> | **Viewpoints** | **Success Rate** |
> |----------------|------------------|
> | M=1            | 0.297 ± 0.029    |
> | M=10           | 0.397 ± 0.019    |
> | M=50           | 0.437 ± 0.044    |
> | M=100          | 0.467 ± 0.015    |
> | M=200          | 0.457 ± 0.022    |
> | M=300          | **0.537 ± 0.029**|
> | M=400          | **0.540 ± 0.035**|
> | M=500          | 0.483 ± 0.019    |
> | M=1000         | 0.493 ± 0.023    |
>
> This result shows that performance improves with a greater number of augmented views, saturating around $M=300$ views.

---

> ### Author Response · Authors · 2025-11-27
> **Result Update - Conclusion and Request for Reconsideration (4/4)**
>
> We thank the reviewer's constructive feedback. The newly added benchmark experiment and multiple ablation studies have substantially strengthened the clarity and principles underlying this paper. **We are confident that this work addresses a novel and important problem while providing a practical, innovative solution with state-of-the-art performance**. Our key contributions include:
>
> 1. **Identifying a critical yet underestimated problem**: We demonstrate that accounting for a manipulation policy's inherent preferences and sensitivities is crucial for successful mobile manipulation. Our experiments reveal that performance decreases dramatically without this consideration (as evidenced by the Reachability baseline), underscoring the necessity of our approach.
> 2. **Proposing a highly practical method with SOTA performance**: Our N2M method not only significantly outperforms geometry-based baselines but also surpasses the only existing policy-aware alternative, Mobi-$\pi$. The extensive benchmarks validate both the effectiveness and novelty of our policy-aware approach.
> 3. **Technical contribution**: We propose a novel viewpoint augmentation method that substantially improves viewpoint robustness and data efficiency while enhancing model generalization. Moreover, it can be seen as a general 3d data processing techniques, which has significant potential applications both within and beyond the domain discussed in the paper.
>
> We are confident that our findings and solutions will advance the field of mobile manipulation. All simulation and real-world implementations (including benchmark code) will be released upon paper acceptance. In light of these comprehensive enhancements and the demonstrated significance of our work, **we respectfully invite our reviewer to reconsider the paper's contributions and impact based on the revised experimental results and manuscript**.

---

> ### Author Response · Authors · 2025-12-02
> **Result Update 2 - add one more quantitative real-world experiment**
>
> Our reviewer requested more quantitative real-world experiments. In response, we designed a multi-stage task (Section 5.3 Comprehensive Case 3). We believe this quantitative experiment plays a crucial role in explaining both the **SOTA performance of our method** and the **importance of the research problem** we address. The details are as follows:
>
> **Task Description**
> We design a multi-stage task (Fig. 14), which requires the robot to: (s1) navigate to a table, (s2) execute a pre-trained grasping policy to pick up an empty chip box, (s3) navigate to a trash bin, and (s4) execute a disposal policy to drop the chip box.
>
> **Experimental settings**
> We evaluate this multi-stage task with 20 trials and under three methods: **Reachability, Human-intuition, and N2M**. The pose of the table, trash bin, and chip box are randomized each trial. Detailed settings are provided in Appendix C.5.
>
> **Results**
> The result is as follows. Note that due to limitations in the rebuttal, we cannot include figures and visual result analysis. We strongly encourage downloading our latest revision to view Section 5.3 Comprehensive Case 3.
>
> | **Method** | Half Task Success(s1+s2) | Full Task Success(s1+s2+s3+s4)  |
> |--------------|----------------------|-----------------------|
> | Reachability | 8/20 (40%) | 3/20 (15%) |
> | Human-intuition | 6/20 (30%) | 5/20 (25%) |
> | N2M | **18/20 (90%)** | **12/20 (60%)** |
>
> **Analysis**
> The result reveals two critical insights:
>
> First, **sole reachability** proves insufficient for effective manipulation. **Even for humans** without prior knowledge of the policy, determining the optimal base pose is challenging. **Notably,** we observed that after several trials, humans begin to discern the policy's positional preferences, gradually achieving higher success rates. **This observation aligns with our design intuition** for N2M, which similarly learns these preferences through trials, or more formally, policy rollouts.
>
> Second, the **sequential nature of this multi-stage task** demonstrates how performance compounds across subtasks, highlighting the importance of optimizing base pose in each subtask. **Our proposed N2M module**, learning from policy rollouts, effectively captures the base preferences, substantially improving both subtask success rates and overall sequence completion. This underscores N2M's significance in multi-task mobile manipulation scenarios where sequential success is paramount.

---

### Author Response · Authors · 2025-12-02
**Brief overview for our Area Chair**

We thank all the reviewers and chairs for taking their valuable time to review this paper. Since the rebuttal contains extensive information, we provide this summary to help quickly understand the situation.

### Paper Overview
- In mobile manipulation, the manipulation policy has strong preferences for initial poses where it is executed. However, the navigation module focuses solely on reaching the task area, without considering which initial pose is preferable for downstream manipulation.
- We identify this critical, yet highly overlooked problem and introduce N2M, a strongly practical solution that guides the robot to a preferable initial pose after reaching the task area, thereby substantially improving task success rates.
- We show state-of-the-art performance of our proposed N2M module with extensive experiments, including 11 experiments across 11 tasks, 1 benchmark, 5 ablation studies, and 1 representation-level visualization analysis.

### Rebuttal Overview

**Reviewer 1**
- **Acknowledgment**: Recognized the performance of N2M and technical innovation on viewpoint augmentation.
- **Concerns**: Questioned whether robot base positioning is already solved by existing methods like B*, suggesting our dramatic improvements might reflect poor baseline implementation rather than genuine advancement. Suggested experiments on basline, ablation, and real-world for stronger support of our claim.
- **Author's Response**: Clarified that base placement remains critical for modern visuomotor policies due to their sensitivity to initial poses. Provided benchmark results comparing N2M against reachability-based methods and Mobi-π, demonstrating significant performance improvements, and added further ablation and real-world experiments to strengthen our claim.

**Reviewer 2**
- **Acknowledgment**: Appreciated the well-structured presentation of technical challenges and solutions. Recognized the performance examined by extensive experiments.
- **Concerns**: Considered base positioning a classic problem with lots of existing solutions, questioning the novelty of our approach compared to prior work.
- **Author's Response**: Demonstrated that previous reachability-based methods fail with modern visuomotor manipulation policies. Explained how the paper completely redesigns the solution, pioneering a new perspective that naturally overcomes previous limitations. Clearly differentiated N2M from the two papers cited by reviewer.

**Reviewer 3**
- **Acknowledgment**: Recognized our work addresses a challenging problem in mobile manipulation with extensive experimental validation.
- **Concerns**: Raised questions about implementation details and suggested further exploration of generalizability.
- **Author's Response**: Provided detailed answers to the reviewer's questions. Explained they have already conducted lots of experiments to demonstrate the generalizability of N2M. But followed reviewer's suggestion, we added a new real-world experiment and also added failure cases to give comprehensive introduction.

**Reviewer 4**
- **Acknowledgment**: Recognized the application value of our work and its advantages in real-time performance, viewpoint robustness, and data efficiency.
- **Concerns**: Noted limited comparisons with alternative methods and questioned how the robot navigates to the predicted pose.
- **Author's Response**: Summarized and analyzed prior work regarding the reviewer’s suggestion to include more discussion and comparison with related methods. Added a detailed quantitative benchmark to significantly strengthen the paper's persuasiveness.

### Final Statement

We strongly assert that **both the problem and our solution are significantly underestimated**. While we deeply appreciate all reviewers' feedback, we believe the initial low scores may stem from unfamiliarity with the latest developments in mobile manipulation. Below, we summarize why our work represents a critical advancement for mobile manipulation systems.

- **Value of Problem is Underestimated**: Prior approaches relying on reachability calculations were sufficient for classical planners but critically fail with modern visuomotor manipulation policies, due to the high sensitivity towards the initial pose. It appears that reviewers are unfamiliar and underestimate the novelty of this challenge, presuming the 'robot base positioning' problem is already solved.
- **Solution is Underestimated**: While all reviewers acknowledge the strong performance of our solution, they have unfortunately taken this success for granted, overlooking the technical novelty required to achieve it. Our newly added benchmark shows N2M significantly surpasses existing methods in both inference speed and success rate by a substantial margin.

We have **absolute confidence** in the innovation and effectiveness of the proposed method. We have addressed all reviewer's concerns and clarified the misunderstandings. Therefore, we respectfully request a reconsideration of our work.

---

### Meta-Review · Area_Chair_pXZ3 · 2025-12-26

**Summary:**

Overall, reviewers agreed the paper targets a practically relevant gap in modular mobile manipulation, i.e., the gap between navigation and manipulation preference. This paper presents a learned N2M module that predicts a multimodal distribution over preferable base poses from point clouds and claims strong data efficiency and broad applicability across tasks. However, reviewers also agreed the paper has relatively limited novelty, and relatively weak baselines. In addition, real-world experiments are conducted mostly by manual rules rather than learned policies, which doesn't justify the original claims. I recommend a rejection to the paper.

**Reviewer Concerns:**

Concerns addressed:
- Benchmark Comparisons: The authors added a quantitative benchmark comparing N2M against reachability-based methods and mobi-$\pi$
- Ablations & statistical reporting: The authors added error bars and several ablations with corresponding error bars
- Lots of clarifications

Concerns that still persist
- Real-world experiment: While one additional real-world experiment was added, the critique that many real-world experiments rely on manual rules still persist. This should be addressed to support the claim of the paper. And we should avoid overclaiming and misleading readers with vague contributions
- Novelty: Even if the proposed learning formulation is reasonable, the paper still does not persuasively differentiate itself from existing lines of work on (i) base placement / positioning policies and (ii) training-free mobile manipulation enablement methods

**Reviewer Scores:**

I think Reviewer tyZd might raise the score to 5, because tyZd focuses more on definition-wise questions, and generalization capabilities, which I believe have been fully addressed. Others might remain the same.

---

### Decision · Program_Chairs · 2026-01-26

Reject